# Predicting the comprehensive geospatial pattern of two ephedrine-type alkaloids for *Ephedra sinica* in Inner Mongolia

Longfei Guo[1], Ping He[1], Yuan He[2], Yu Gao[1], Xiaoting Zhang[1], Tongtong Huo[1], Cheng Peng[3], Fanyun Meng[1] *

1 Beijing Key Laboratory of Traditional Chinese Medicine Protection and Utilization, Faculty of Geographical Science, Beijing Normal University, Beijing, China, 2 State Key Laboratory of Earth Surface Processes, Faculty of Geographical Science, Beijing Normal University, Beijing, China, 3 School of Pharmacy, Chengdu University of Traditional Chinese Medicine, Chengdu, China

* mfy@bnu.edu.cn

**Data Availability Statement:** Supplementary data to this article can be found online at https://doi.org/10.5061/dryad.8cz8w9gsk.

## Abstract

*Ephedra sinica* Stapf. is a shrubby plant widely used in traditional Chinese medicine due to its high level of medicinal value, thus, it is in high demand. Ephedrine (E) and pseudoephedrine (PE) are key medicinal components and quality indicators for *E. sinica*. These two ephedrine-type alkaloids are basic elements that exert the medicinal effect of *E. sinica*. Recently, indiscriminate destruction and grassland desertification have caused the quantity and quality of these pharmacological plants to degenerate. Predicting potentially suitable habitat for high-quality *E. sinica* is essential for its future conservation and domestication. In this study, MaxEnt software was utilized to map suitable habitats for *E. sinica* in Inner Mongolia based on occurrence data and a set of variables related to climate, soil, topography and human impact. The model parametrization was optimized by evaluating alternative combinations of feature classes and values of the regularization multiplier. Second, a geospatial quality model was fitted to relate E and PE contents to the same environmental variables and to predict their spatial patterns across the study area. Outputs from the two models were finally coupled to map areas predicted to have both suitable conditions for *E. sinica* and high alkaloid content. Our results indicate that *E. sinica* with high-quality E content was mainly distributed in the Horqin, Ulan Butong and Wulanchabu grasslands. *E. sinica* with high-quality PE content was primarily found in the Ordos, Wulanchabu and Ulan Butong grasslands. This study provides scientific information for the protection and sustainable utilization of *E. sinica*. It can also help to control and prevent desertification in Inner Mongolia.

## 1. Introduction

Globally, medicinal plant diversity is critical to human health [1]. However, traditional Chinese medicine (TCM) resources, which are irreplaceable, have experienced unsustainable development in recent decades due to their high market value [2]. Many Chinese medicinal plants have declined in quantity and quality as a result of global climate change and human activities, with some plants on the verge of extinction [3]. *Ephedra sinica* Stapf is a valuable

**Funding:** Funding for our research was provided by the National Natural Science Foundation of China (Grant No. 81072999) and the open Research Fund of Chengdu University of Traditional Chinese Medicine State Key Laboratory of Southwestern Chinese Medicine Resource (No. SKLTCM2022012). The funders had no role in study design, data collection and analysis, decision to publish, or preparation of the manuscript.

**Competing interests:** The authors have declared that no competing interests exist.

medicinal resource that is mainly distributed in the Inner Mongolia Autonomous Region, China [4]. The medicinal components of this plant are known for favoring sweating, alleviating colds, promoting lung health, relieving asthma, and stimulating detumescence. It is commonly used in TCM to treat a variety of ailments, including bronchitis, asthma, whooping cough, colds and obesity [5]. Alkaloids are the most effective chemical components of *E. sinica*, and ephedrine hydrochloride (E) and pseudoephedrine hydrochloride (PE) are important components for exerting the pharmacological effects of *E. sinica*. As a result, E and PE contents are used as marker compounds for quality control [6, 7]. E is an effective component in providing analgesia services [8], regulating blood pressure [9], stimulating the central nervous system [10], and relaxing bronchial smooth muscles [11], along with other beneficial effects. PE is an effective component with diuretic [12] and anti-inflammatory properties [13]. Previous studies have shown that ephedrine-type alkaloids differ in content and quality among species and geographical locations [14]. Furthermore, *E. sinica* is a shrubby plant with windbreak, sand fixation, soil erosion prevention, grassland protection and sandy land ecology improvement functions. It has high ecological value in arid and semiarid environments [15]. However, the increasing international market demand for natural alkaloid contents has led to overharvesting of *E. sinica*. In the meanwhile, due to grassland desertification in Inner Mongolia, *E. sinica* has declined dramatically in recent years and is even endangered in some regions. Predicting suitable habitat for this plant can benefit its conservation and restoration.

SDMs (species distribution models) are widely used in geography, ecology, and species conservation and are effective tools for studying species and environmental issues, particularly in the fields of endangered species protection and environmental impact assessment [16, 17]. Various SDM algorithms, including CLIMEX, DOMAIN, the genetic algorithm for rule set production (GARP), BIOMOD-2 and maximum entropy (MaxEnt), have been used to assess ecological requirements and predict habitat suitability for medicinal plant species [18–20]. MaxEnt is frequently chosen since it is simple to operate and performs well with a small number of samples [21–23]. For example, based on environmental data for current and future climate scenarios, Zhang et al. [24] used MaxEnt to model present and future suitable habitats for two peony species (*Paeonia delavayi* and *Paeonia rockii*), evaluating the importance of environmental factors in shaping species distributions and identifying distribution shifts under climate change scenarios. Gaikwad et al. [25] used MaxEnt to predict the potential distribution of 431 indigenous medicinal plant species in Australia based on bioclimatic variables. Most current studies, however, have ignored the chemical components that influence the quality of medicinal plants.

The impact of the environment on medicinal materials is reflected in the suitability of species growth and spatial differences in medicinal plant quality [26]. Medicinal plant habitat provides the foundation for producing and accumulating effective components, but the environmental variables that influence these components may differ from those influencing habitat suitability. Some of the effective components of pharmacological plants are those that allow plants to survive and resist adversity [27]. Areas hosting suitable conditions for high-quality populations of a pharmacological plant species (i.e., populations whose individuals show high content levels of the plant's medicinal components) may not completely match areas where the species actually occurs. Thus, studies on high-quality suitable habitat for TCM plants should pay attention not only on the suitable habitat but also on the geospatial pattern of the effective medicinal components. Currently, many researchers are increasingly focused on the divergence of quality in effective medicinal contents among different regions [6, 28, 29]. Li et al. [30] predicted the distributions of three *Coptis* herbs (*Coptis chinensis* Franch., *Coptis deltoidea* C. Y. Cheng et Hsiao and *Coptis teeta* Wall) with the MaxEnt model and analyzed their corresponding chemical compositions by high-performance liquid chromatography (HPLC). They found that climate variables had a significant impact on habitat suitability

and alkaloid contents in three *Coptis* herbs. Guo et al. [29] used a species distribution model and a component quality model to map the spatial pattern of *Ophiocordyceps sinensis* quality and also investigated how environmental factors affected the distribution and adenosine content pattern of *O. sinensis*. Furthermore, a study on the influence of habitat on the baicalin content in *Scutellaria baicalensis* Georgi revealed that climatic factors were important indicators of both the potential distribution and baicalin production of the plant. There is currently no such spatial pattern analysis for *E. sinica*.

In this study, our goal was to predict the comprehensive geospatial pattern of *E. sinica* by using E and PE as quality indicators. We first used the MaxEnt model and occurrence records to simulate the potential distribution of *E. sinica*. Then we used a geospatial quality model and ephedrine-type alkaloid content data to predict the spatial patterns of E and PE contents. Finally, based on the comprehensive geospatial quality model, we obtained the comprehensive geospatial pattern of *E. sinica*. The results will help determine which environmental factors influence the distribution of suitable habitats for *E. sinica* and the effective production of ephedrine-type alkaloids. Our research will provide theoretical support for the conservation and sustainable utilization of *E. sinica* in Inner Mongolia and afford a systematic approach for predicting the production zoning of natural medicinal materials.

## 2. Materials and methods

### 2.1. Study area and occurrence data of Ephedra sinica

*E. sinica* is primarily distributed in the Inner Mongolia Autonomous Region of China (37˚ 24′~53˚23′N, 97˚12′~126˚04′E) [31]. Inner Mongolia has a diverse range of wild medicinal resources, including over 1000 species of medicinal plants, 120 species of medicinal animals, and 40 species of medicinal minerals. In our research, Inner Mongolia was artificially divided into three parts: east, central and west. The grasslands are mostly distributed in eastern and central Inner Mongolia, including Hulun Buir, Uragai, Xilingol, Horqin, Ulan Butong, Wulanchabu and Ordos, while the western region is arid and includes the Alxa desert.

Occurrence records of *E. sinica* and sample sites with content data for two target ephedrine-type alkaloids were obtained through published literature [14, 28, 32] and the Chinese Virtual Herbarium databases (CVH, https://www.cvh.ac.cn). The coordinates were picked via Google Earth (http://ditu.google.cn/) when distribution and quality data lacked precise latitude and longitude. To reduce the potential detrimental effect of spatial autocorrelation in the training dataset, duplicate records were removed from the dataset, and only one record was retained within each grid cell (1 km × 1 km).

Ultimately, 78 occurrence records and 35 sampling sites with two ephedrine-type alkaloids in *E. sinica* were analyzed (Fig 1).

### 2.2. Environmental data

Because environmental variables contribute to determining the simulation results of models [33], the selection of environmental variables should take into account not only data availability but also biological correlation [34, 35]. *E. sinica* grows in sandy soil with good air permeability and is mainly found on slopes and flat land in arid or semiarid areas. Based on the distribution characteristics of *E. sinica*, we initially selected four groups of environmental variables: climatic, soil, topographic and human activity. Climatic variables represented the average for the years 1970–2000 and comprised solar radiation, vapor pressure and bioclimatic variables derived from monthly temperature and precipitation [36]. Climatic variables were obtained from the WorldClim Database (www.worldclim.org) [37]. The data received from WorldClim can be used for mapping and spatial modeling [38–40]. Soil variables included soil

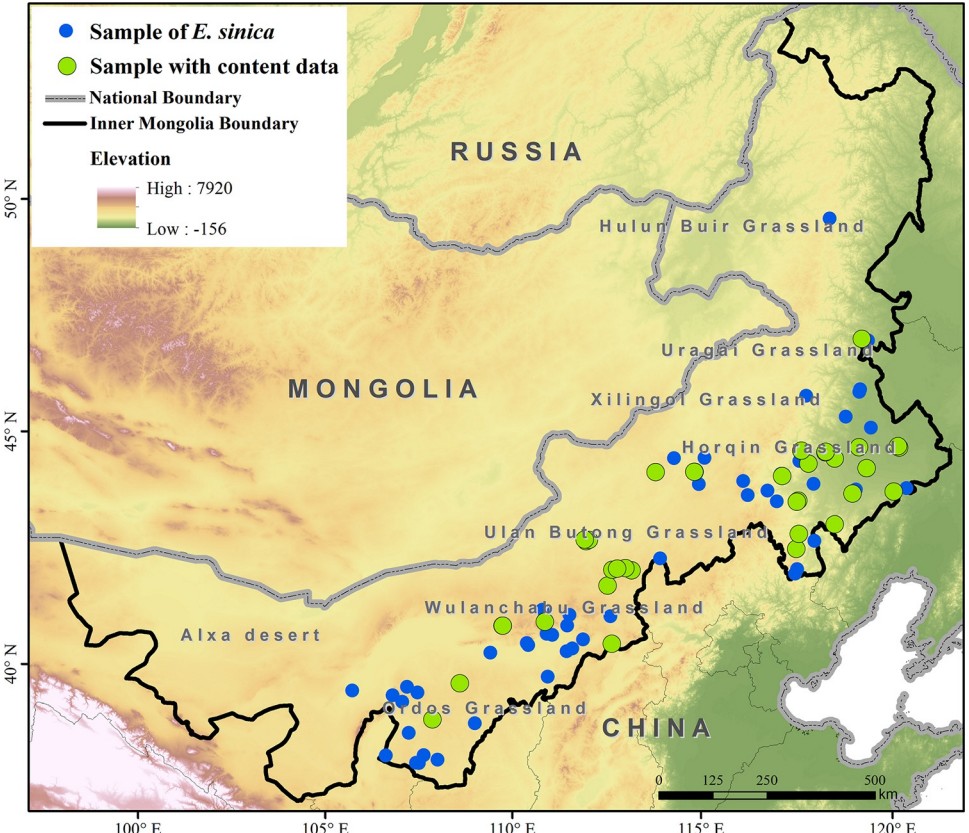

**Fig 1. Study area and sampling site distribution.** The boundary was obtained from Natural Earth (http://www.naturalearthdata.com) and was further processed using ArcGIS version 10.3 software.

types from the Resource and Environment Science and Data Center (https://www.resdc.cn/Default.aspx), soil particle-size distribution dataset [41] and soil quality data from the Harmonized World Soil Database (HWSD) (https://www.fao.org/soils-portal/data-hub/soil-maps-and-databases/harmonized-world-soil-database-v12/en). Topographic variables included elevation, slope and aspect. Slope and aspect data were generated from elevation data obtained from WorldClim using the spatial analyst tool of ArcGIS 10.3 software (ESRI, Redlands, CA, USA). Human activity variables included human footprint [42], population density [43] and global human modification of terrestrial systems [44], which were derived from the Socioeconomic Data and Applications Center (http://sedac.ciesin.columbia.edu). All environmental variables had a 30-s (approximately 1 km$^2$) spatial resolution.

Multicollinearity among the environmental variables will increase the uncertainty of the model results [45, 46]. To avoid model overfitting, we used the Pearson correlation coefficient (r) to test the correlation between the environmental variables in the four groups and ensure that all pairwise |r| values were less than 0.75 [47]. Consequently, only 17 variables were chosen for the next step after removing highly correlated environmental variables (Table 1).

## 2.3. Model implementation and evaluation

By coupling a species distribution model with a geospatial quality model (GQM), we developed a comprehensive geospatial quality model (CGQM) to predict the geospatial pattern of two ephedrine-type alkaloids in *E. sinica* (Fig 2).

**Table 1. Environmental variables used or not used in model, along with the respective relative contribution scores from the fitted MaxEnt model.**

| Variable type | Code(Unit) | Description | Variables used in modeling | Contribution (%) |
|---|---|---|---|---|
| Climatic variables | bio1(˚C) | Annual mean air temperature | √ | 0.3 |
| | bio2(˚C) | Mean diurnal temperature range (max. temp–min. temp) | √ | 0.2 |
| | bio3 | Isothermality (Bio2/Bio7) × 100 | √ | 4.3 |
| | bio4(˚C) | Temperature seasonality | | |
| | bio5(˚C) | Max temperature of warmest month | | |
| | bio6(˚C) | Min temperature of coldest month | | |
| | bio7(˚C) | Temperature annual range | √ | 29.1 |
| | bio8(˚C) | Mean temperature of wettest quarter | | |
| | bio9(˚C) | Mean temperature of driest quarter | | |
| | bio10(˚C) | Mean temperature of warmest quarter | | |
| | bio11(˚C) | Mean temperature of coldest quarter | | |
| | bio12(mm) | Annual precipitation | √ | 7.4 |
| | bio13(mm) | Precipitation of wettest month | | |
| | bio14(mm) | Precipitation of driest month | | |
| | bio15(%) | Coefficient of variation of precipitation | √ | 17.3 |
| | bio16(mm) | Precipitation of wettest quarter | | |
| | bio17(mm) | Precipitation of the driest quarter | | |
| | bio18(mm) | Precipitation of warmest quarter | | |
| | bio19(mm) | Precipitation of coldest quarter | | |
| | srad(kJ·m$^{-2}$·d$^{-1}$) | Solar radiation | √ | 9.6 |
| | vapr(hPa) | Vapour pressure | √ | 0.6 |
| Soil variables | soil | Soil type | √ | 2 |
| | clay1 | Topsoil Clay Fraction (0–30cm) | | |
| | clay2 | Subsoil Clay Fraction (30–100cm) | | |
| | sand1 | Topsoil Sand Fraction (0–30cm) | √ | 1.3 |
| | sand2 | Subsoil Sand Fraction (30–100cm) | | |
| | sq1 | Nutrient availability | √ | 0.4 |
| | sq2 | Nutrient retention capacity | | |
| | sq3 | Rooting conditions | √ | 0.9 |
| | sq4 | Oxygen availability to roots | | |
| | sq5 | Excess salts | √ | 0.7 |
| | sq6 | Toxicity | √ | 0.1 |
| | sq7 | Workability (constraining field management) | | |
| Topographical variables | ele(m) | Elevation above sea level | √ | 3.6 |
| | slop(%) | Slope | | |
| | asp(degrees) | Aspect | √ | 1.1 |
| Human activity variables | hf | Human Footprint | | |
| | den | Population Density | | |
| | ter | Global Human Modification of Terrestrial Systems | √ | 21.6 |

Note:

"√" represents environmental variables that are used during model execution.

**2.3.1. Distribution model.** MaxEnt, one of the most successful SDMs in predicting species distribution [48], is based on the maximum entropy principle and Bayesian estimates [33]. We used MaxEnt (Version 3.4.0; http://www.cs.princeton.edu/eschapire/MaxEnt/) to project the potential distribution of *E. sinica* across Inner Mongolia [49]. Feature classes (FC) and the

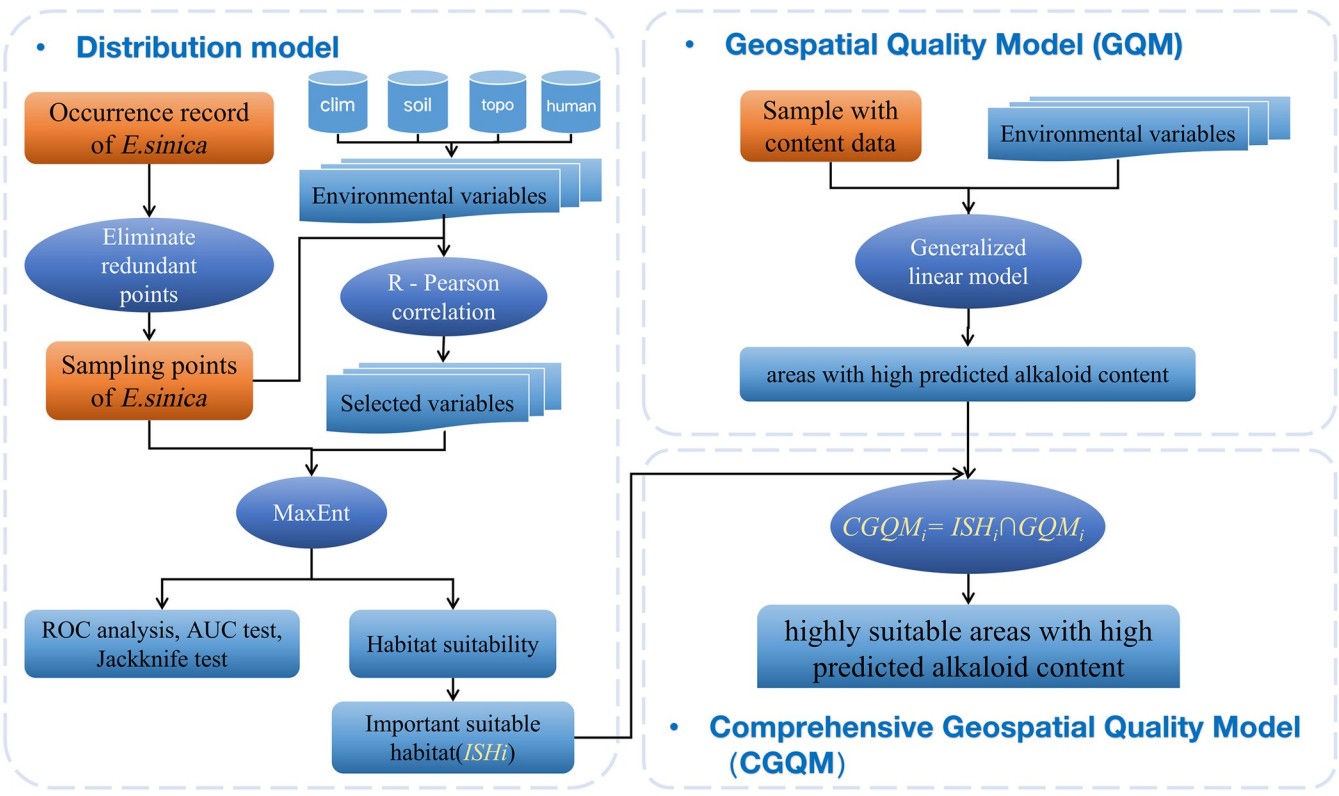

**Fig 2. Schematic representation of the implemented methodological approaches.**

regularization multiplier (RM) influence the accuracy of the MaxEnt model [50, 51]. Thus, we used the 'kuenm' package in R software to select the most appropriate parameters for MaxEnt to simulate the potential distribution of *E. sinica* [52]. In our study, 75% of the occurrence data were randomly selected to fit the model, and the remaining data were used to test the model. In total, 248 candidate models were tested by combining RM values ranging from 0.5 to 4.0 (increments of 0.5, total 8 values) with all 31 combinations of the five FC settings (l, q, p, t, h; where l = linear, q = quadratic h = hinge, p = product and t = threshold) [53]. Model selection was based on (i) omission rates (E ≤ 0.05), which represent the proportion of incorrectly predicted test records by the model, and (ii) Delta Akaike Information Criterion (dAIC < 2), which indicates the models with the best trade-offs between data fitting and complexity [54]. The dAIC was the difference between the AIC of a particular model and the lowest AIC of all candidate models [55]. Finally, the selected MaxEnt model was fitted using linear and threshold features (i.e., FC = lt), with RM = 3.5. The maximum number of iterations was set to 1000, the logistic output format was set to logistic, and 10 replicated runs of cross-validation were used to reduce the uncertainty of the model. The model generated *E. sinica* suitability prediction, which we transformed into a raster with values ranging from 0 to 1 for further analysis. The final predicted suitability values were grouped via natural breaks into four classes of High, Medium, Low and Unsuitable.

The receiver operating characteristic (ROC) curve and area under the curve (AUC) were used to verify and evaluate the accuracy and robustness of the model. AUC values range from 0 to 1, with values lower than 0.5 indicating that model predictions are not better than random ones and 1 representing perfect discrimination. When the AUC value is above 0.8, the model result is satisfactory, and when the AUC value is between 0.9 and 1, the model result is

excellent [56, 57]. Furthermore, we used a jackknife test to determine the relative importance of the explanatory variables.

**2.3.2. Geospatial quality model.**  In this study, we built a GLM as a geospatial quality model to simulate the spatial pattern of two ephedrine-type alkaloids, as shown in Eq (1) [58]

$$g[E(Y)] = LP = \alpha + X\beta \tag{1}$$

In the GLM, the linear predictor (LP) is combined with the predictor variables $X_p(p = 1,2,\ldots,j)$, where $\alpha$ is the intercept, $\beta$ is the vector of the regression coefficients, $\mu = E(Y)$ denotes the conditional expectation of Y given $X = $    , and g() denotes the link function; here, we chose the Gaussian type. The corresponding terms for the *i*th observation in the sample are as follows:

$$g(\mu_i) = \alpha + \beta_1 \quad _{i1} + \beta_2 \quad _{i2} + \cdots + \beta_j \quad _{ij} \tag{2}$$

We used SPSS (IBM SPSS Statistics 26; https://www.ibm.com/cn-zh/analytics/spss-statistics-software) to perform a stepwise regression to refine the initial GLM, thus maintaining only the informative predictors [29]. The F-statistic and R-squared were used to validate the results of the spatial quality model, which were calculated with Eq (3) and Eq (4):

$$F = \frac{R^2/p}{(1 - R^2)/(n - p - 1)} \tag{3}$$

$$R^2 = 1 - \frac{\sum_{i=1}^{n}(y_i - \hat{y}_i)}{\sum_{i=1}^{n}(y_i - \bar{y})} \tag{4}$$

where n is the number of sample sites, $p$ is the number of predictive variables in the model, and $(n-p-1)$ denotes the degrees of freedom.

**2.3.3. Comprehensive geospatial quality model.**  To evaluate the geospatial quality of the two ephedrine alkaloids in Inner Mongolia, it is necessary to first determine the growth and distribution area of the species and then analyze the alkaloid content according to habitat suitability. From this process, we identified medium- and high-suitability habitats as important suitable habitats (ISH). A comprehensive geospatial quality model is then constructed [Eq (5)],

$$CGQM_i = ISH_i \cap GQM_i \tag{5}$$

where $CGQM_i$ is the output of the comprehensive geospatial quality model for the two ephedrine-type alkaloids of *E. sinica* in the *i*th evaluation unit, $ISH_i$ is the important suitable habitat simulation result of *E. sinica* in the *i*th unit, and $GQM_i$ is the *i*th unit simulation result of spatial quality.

## 3. Results

The MaxEnt-based species distribution model (SDM) and regression-based geospatial quality model served as the foundation for the comprehensive geospatial quality model. We will discuss these two base models separately.

## 3.1. Predicted suitable habitat model for Ephedra sinica and its accuracy

The selected model presented an omission rate of 0.05, dAIC of 0, and AUC value of 0.965, indicating that the model performed well and was reliable in discriminating suitable areas for *E. sinica*. The predicted suitability was divided into four levels: high suitability ($> 0.55$) was 161400 km$^2$, medium suitability (0.34–0.56) was 260400 km$^2$, low suitability (0.14–0.34) was

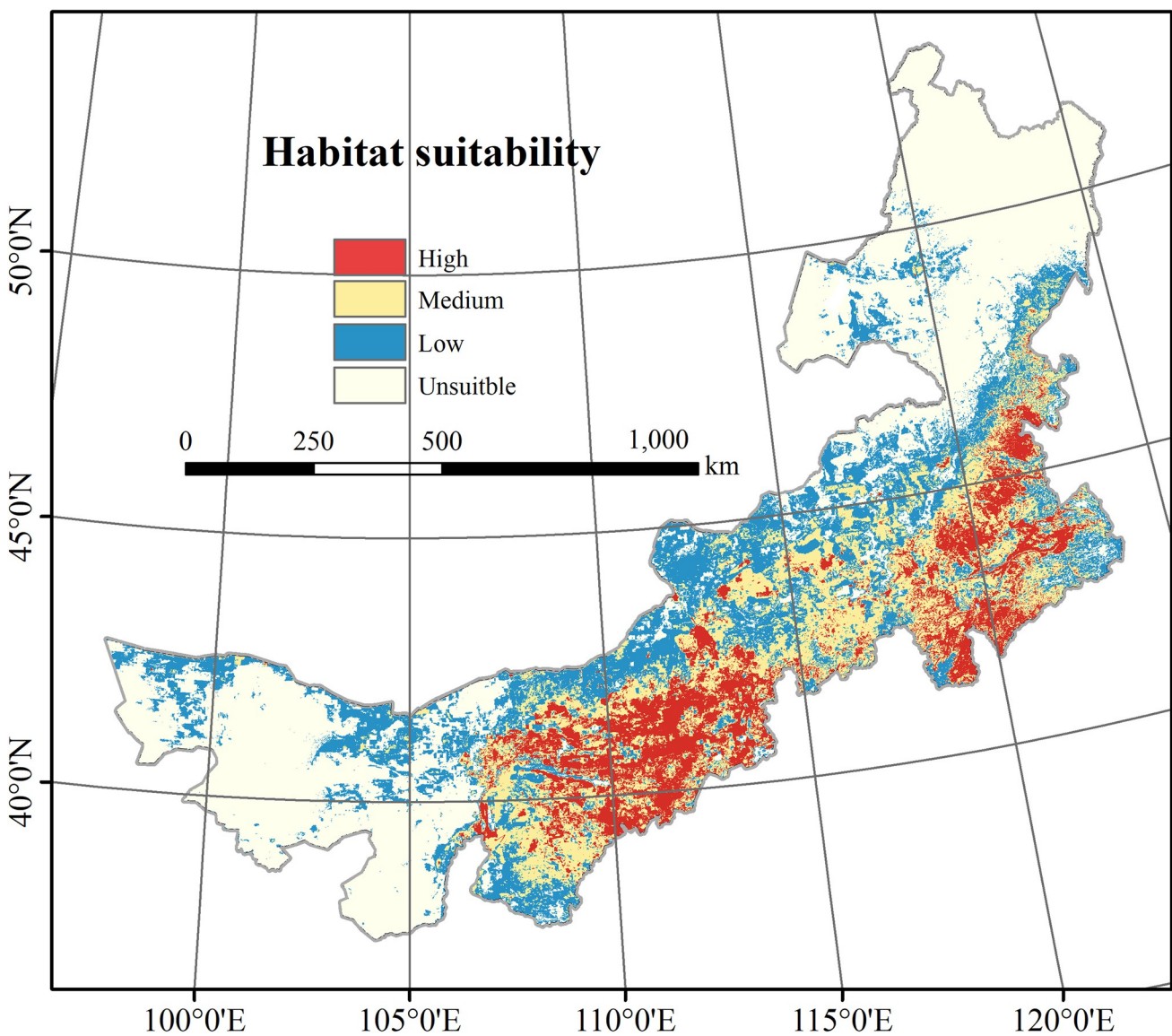

**Fig 3. Predicted potential habitats for *E. sinica*.** This map was made in ArcGIS 10.3 using the resulting raster produced by MaxEnt. The original boundary was obtained from Natural Earth (http://www.naturalearthdata.com).

285700 km$^2$, and the remaining area was unsuitable (<0.13) (Fig 3). High-suitability habitat was mainly distributed in the Horqin, Wulanchabu and Ordos grasslands. Medium-suitability habitat mainly was found around high-suitability habitat in the Horqin, Wulanchabu, Ordos and Ulan Butong grasslands. ISH covered approximately ca. 421800 km$^2$.

The internal jackknife test of factor importance showed that bio7 (temperature annual range, accounting for 29.1%) was the most important influencing factor, followed by ter (global human modification of terrestrial systems, 21.6%), bio15 (coefficient of variation of precipitation, 17.3%), srad (solar radiation, 9.6%) and bio12 (annual precipitation, 7.4%) (Table 1); their cumulative contribution reached 85%. The variable partial response curves reported in Fig 4 show how MaxEnt predictions of habitat suitability for *E. sinica* changed along the gradient of the most influential variables [30]. Moreover, our model results indicated that the suitable range for bio7 was approximately 45˚C to 50˚C, and the optimal value of bio7

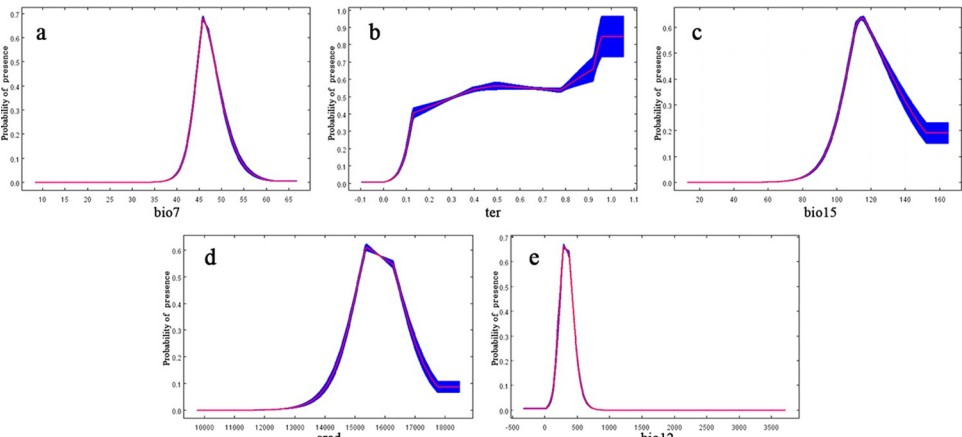

**Fig 4. Response curves for important environmental predictors in the species distribution model for *E.* sinica.**

was approximately 46°C. In addition, bio15 ranged from 105% to 130% with an optimal value at approximately 115%; srad ranged from 15000 kJ·m$^{-2}$·d$^{-1}$ to 16500 kJ·m$^{-2}$·d$^{-1}$, with the highest predicted suitability value at 15500 kJ·m$^{-2}$·d$^{-1}$; bio 12 ranged from 200 mm to 450 mm with an optimal elevation at approximately 270 mm. Furthermore, as the value of ter increased above 0.1, an increase in the probability of presence was observed. Such a suitable range of environmental variables was more common in the ISH area.

## 3.2. Predicted spatial patterns of two ephedrine-type alkaloids and their accuracy values

We used the geospatial quality model to predict the geospatial pattern of E and PE content. The relationship between E content and the environmental variables was $Y_1 = 3.255+-0.193X_1+0.001X_2$, where $Y_1$ is the E content, $X_1$ is bio2 (mean diurnal temperature range), and $X_2$ is aspect. We applied a significance test to the model. The observed value of the F statistic was 6.182, the P value was 0.006, and $R^2$ was 0.279. This result suggests that the model explained a nonnegligible portion of the spatial variability in E content, and the key variable of E content accumulation was the mean diurnal temperature range. The spatial distribution of the E content in the different regions is shown in Fig 5A.

The geospatial quality model of PE content was represented by the equation $Y_2 = 3.792-- 0.159X_1$, where $Y_2$ is the PE content and $X_1$ is bio8 (mean temperature of the wettest quarter). The observed value of the F statistic was 22.747, the P value was 0.000, and $R^2$ was 0.408. This result demonstrated that the model has noticeable explanatory power. Bio8 is the most important variable for PE content, and the projection of PE content across Inner Mongolia resulting from the corresponding geospatial quality model is shown in Fig 6A.

Climate, particularly temperature patterns, is apparently the main factor determining the accumulation of E and PE contents in *E. sinica* samples because bio2 and bio8 emerged as the main variables influencing the spatial distribution of E and PE contents.

## 3.3. Comprehensive spatial pattern of alkaloids in Ephedra sinica

We used ISH as a constraint to restrict the predictions of E and PE contents to areas with high suitability for *E. sinica*. According to the CGQM, areas with high-quality E content (i.e., > 0.8%) were found primarily in the Horqin, Ulan Butong and Wulanchabu grasslands (Fig 5B),

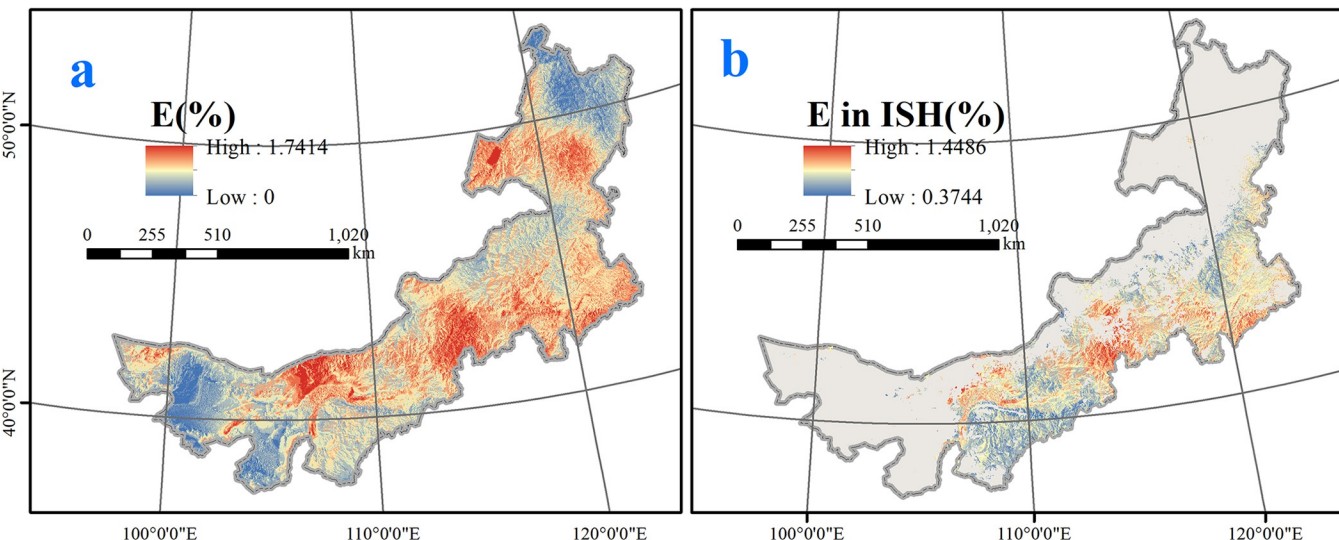

**Fig 5.** a. E content across Inner Mongolia; b. E content in important suitable habitats for *E. sinica*. This map was made in ArcGIS 10.3 using the resulting rasters produced by the geospatial quality mode and distribution model. The original boundary was obtained from Natural Earth (http://www.naturalearthdata. com).

and they covered approximately 138600 km². To some extent, the E content showed distribution characteristics of high quality in the east and low quality in the west.

Areas with high-quality PE content (i.e., > 0.8%) resulting from the CGQM mainly spanned the Ordos, Wulanchabu and Ulan Butong grasslands, whereas the PE content distributed in the Horqin grassland in eastern Inner Mongolia is low (Fig 6B). The cumulative area of *E. sinica* with high-quality PE content was approximately 146400 km². This result means that the PE content decreases from west to east.

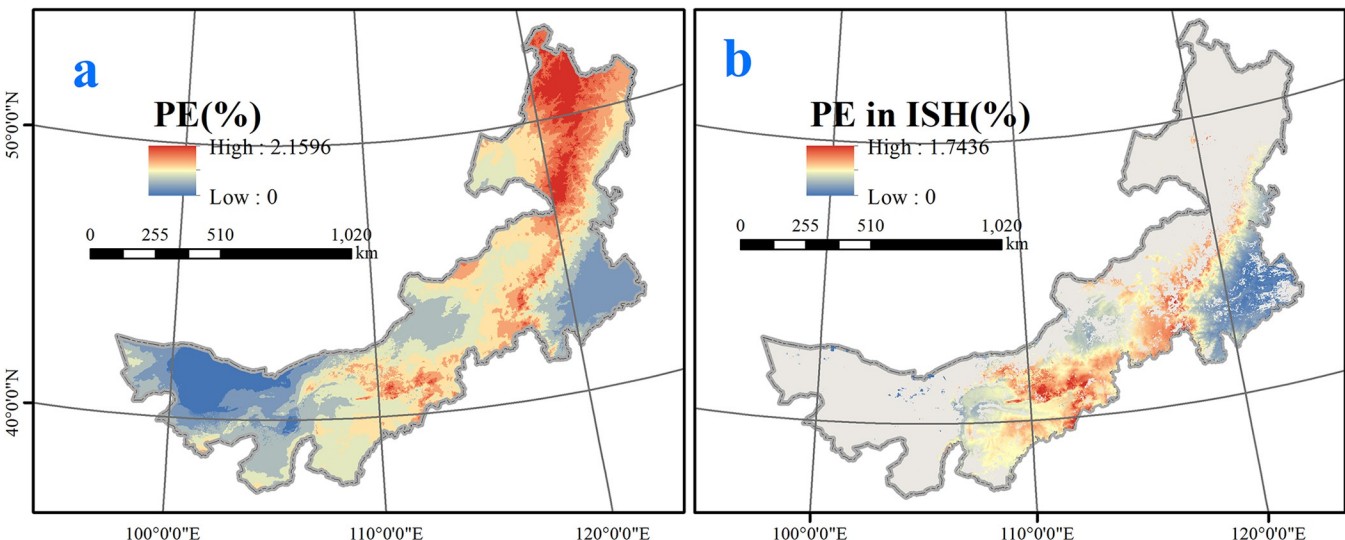

**Fig 6.** a. PE content across Inner Mongolia; b. PE content in important suitable habitats for *E. sinica*. This map was made in ArcGIS 10.3 using the resulting rasters produced by the geospatial quality mode and distribution model. The original boundary was obtained from Natural Earth (http://www.naturalearthdata. com).

## 4. Discussion

Overharvesting and overgrazing have resulted in the destruction of suitable habitat for *E. sinica*, and the quality of *E. sinica* has also declined due to environmental degradation and the influence of human activities [59, 60]. Therefore, more emphasis must be placed on the conservation and efficient utilization of this species.

This study used a MaxEnt model to simulate the potential distribution of *E. sinica* in Inner Mongolia, for which the AUC value was 0.965, indicating excellent discrimination performance. Furthermore, we used a stepwise regression-based model to predict, across Inner Mongolia, the content of two ephedra-type alkaloids characterizing *E. sinica*, and the F-statistic and R-squared for the geospatial quality model results performed well. We obtained the final comprehensive geospatial distribution for two ephedrine-type alkaloids of *E. sinica* by coupling the results of these two models. The findings will promote the sustainable use and appropriate conservation of *E. sinica* in Inner Mongolia by indicating which areas can potentially host larger *E. sinica* populations with high-quality ephedrine-type alkaloid content; these conditions can increase harvesting efficiency. Our results also make an important contribution to the enhancement of medicinal value and protection of biodiversity in Inner Mongolia.

### 4.1. Model application and optimization

Simulating the geospatial pattern of the contents for effective medicinal plant components faces two challenges: limited data and model selection [61]. We needed to collect not only occurrence records of *E. sinica* but also sites with corresponding effective component data; however, collecting data on components is difficult because few studies have been done on the spatial patterns of the two ephedrine-type alkaloids [62]. We believe that component data richness could be increased by using field work to collect the medicinal parts of the target plant species and by implementing additional laboratory work to measure the effective components if corresponding hardware devices are available. As simulating the geospatial pattern of effective components is related to actual production activities, the model should be simple to operate and easy to understand, and we need to intuitively understand the relationship between environmental variables, plant distribution and metabolite contents. In our research, the MaxEnt model was adopted to predict the distribution of *E. sinica* in Inner Mongolia, and a GLM was used to predict the geospatial pattern of two ephedrine-type alkaloids. According to the chosen evaluation metrics, both models performed well, and they identified climate, human modifications and topography as key environmental factors affecting the distribution of *E. sinica* and the spatial pattern of alkaloid content. The fact that we used environmental predictors with approximately 1 $km^2$ resolution may facilitate the development of general guidelines on the conservation and management of *E. sinica* in Inner Mongolia at a relatively fine spatial scale [63].

### 4.2. Effects of environmental variables on Ephedra sinica

The growth and distribution of *Ephedra* and protection of ephedrine-type alkaloids are the result of a complex interplay of various ecological factors, primarily climate, topography, soil and human properties [64].

*Ephedra* grows in arid environments, and climate factors are the most important driver for predicting the *Ephedra* distribution range. Precipitation is the key variable for predicting the distribution of *E. foliata*, and temperature seasonality is the most influential bioclimatic factor of *E. gerardiana* [65]. Precipitation and temperature are also important factors in the growth of *E. sinica*, *E. intermedia* and *E. equisetina* [35]. In addition, climate emerged as the main factor driving the distribution of *Rosa arabica* Crep., which frequently cooccurs with *Ephedra*

species [22]. Similar to previous studies, our results indicated that climate-related predictors were the main variables affecting habitat suitability for *E. sinica*. The major climate variables predicting the distribution of *E. sinica* were temperature annual range (29.1%), global human modification of terrestrial systems (21.6%), coefficient of variation of precipitation (17.3%), solar radiation (9.6%) and annual precipitation (7.4%). Because *E. sinica* is a perennial plant whose perennial roots persist in the soil for many years, annual climatic factors are more likely to affect its growth and distribution. Annual temperature range helps determine the appropriate thermal conditions that *E. sinica* populations in Inner Mongolia need to grow, while precipitation provides adequate water to alleviate soil drought, supply nitrogen fertilizer to the soil via nitrogen compounds in snow meltwater, and moisten the underground roots to aid perennial plant germination in the following spring [66]. Temperature and precipitation play key roles in plant survival, especially for species distributed in arid areas where living conditions are harsh [23, 67]. In addition, solar radiation can directly regulate the photosynthesis and respiration of plants, as well as influence production and reproduction [62]. Furthermore, global human modification of terrestrial systems (23%) is an important variable for the distribution of *E. sinica* because *E. sinica* grows primarily in grasslands, where farming and animal husbandry activities and cultivation behavior change the land cover. However, soil variables, such as soil texture, vertical soil properties and root infiltration conditions, directly affect the survival and growth of perennial roots, thus indicating that soil properties are important drivers of *E. sinica* distribution at finer spatial scales.

Secondary metabolites from plants are among the main sources of pharmaceuticals, food additives, flavors, and other industrial materials [68]. Many plants are able to overcome and adapt to environmental stress by generating secondary metabolites, which are important indicators for assessing the quality of traditional Chinese medicine resources [69]. Temperature, humidity, light intensity, and the availability of water and soil all have an impact on plant physiological activity and secondary metabolite production [70]. Ephedrine-type alkaloids, which are the main secondary metabolites of *Ephedra*, are important for estimating the quality of crude drugs and safe medication [71]. Temperature is an important factor affecting the accumulation of alkaloids, as it leads to different trends. To cope with heat stress, plants implement various mechanisms that are regulated at the molecular level, which can improve plant heat stress tolerance and enable plants to thrive under heat stress. Some scholars have found that peramine and Lolitrem B in perennial ryegrass reach maximum accumulation under high temperatures in summer [72]. Furthermore, different alkaloids exhibit different change trends when subjected to different temperature treatments [73]; for instance, the concentrations of ergine, ergonovine, and total ergot alkaloids are significantly higher when *Festuca sinensis* is maintained under short-term cold stress [74]. In our study, the mean diurnal temperature range was the key variable for the generation of ephedrine (E), and it was positively correlated with the change in E content. When the temperature rose, the E content increased. The most important factor for the production of pseudoephedrine (PE) was the mean temperature of the wettest quarter, which was negatively correlated with PE content, so that the latter increased with lower temperature in the wettest season. Temperature influences photosynthesis, respiration, water relations and membrane stability, as well as hormone, primary and secondary metabolite levels in *E. sinica* [75]. Simultaneously, ephedrine-type alkaloids are important organic osmotic regulators, and precipitation has a crucial impact on their production [76].

To summarize, because *E. sinica* and its ephedrine-type alkaloids are sensitive to different environmental variables, it is necessary to consider local characteristics when protecting and using *E. sinica*. According to our study, this species is predicted to be distributed in the main grasslands of Inner Mongolia. The survival and competition potential of local populations of

*E. sinica* should be evaluated carefully, and the destruction of local habitats and biodiversity should be avoided in the process of artificial cultivation [63].

## 4.3. Different applications of Ephedra sinica based on regional comprehensive quality

The spatial distribution patterns of the two ephedrine-type alkaloid components of *E. sinica* revealed distinct trends [77]. Previous research showed that the geospatial patterns of E content for *E. sinica* decreased from east to west in Inner Mongolia, China, but the PE content experienced the opposite trend [14]. However, some researchers argue that the diversity in E and PE contents cannot be explained by the differences in longitude among the sample distribution areas [28]. According to our comprehensive spatial pattern results, two ephedrine-type alkaloids differed from east to west in Inner Mongolia. The E content was high in the east and low in the west, and the PE gradually became equal to or even higher than the E content from east to west. At the same time, the geospatial distribution of the two ephedrine-type alkaloid contents showed a negative correlation, exemplified by the high E content and low PE content in the Horqin grassland. In contrast, PE content was high, and E content was low in the Ordos grassland.

E and PE contents are important components for *E. sinica* medicinal value [78]. Because the functions of the two ephedrine-type alkaloids differ, their applications differ. As a result, the extraction zones of E and PE should be planned. Reasonable and scientific utilization should be carried out based on the difference in geospatial patterns for E and PE contents. For example, *E. sinica* with high E content should be collected in the Horqin, Ulan Butong and Wulanchabu grasslands. While places with high PE content could be used for extraction of PE, such as in the Ordos, Wulanchabu, and Ulan Butong grasslands. This kind of planning is advantageous for ephedrine-type alkaloids production planning in different regions [79].

## 4.4. Conservation strategies for Ephedra sinica

As a result of rapidly changing climate and human-caused ecological destruction, the distribution of *E. sinica* in Inner Mongolia has shifted from widespread to fragmented distribution in recent decades [80, 81]. Meanwhile, because of the high medicinal value of *E. sinica*, which is widely used in both traditional Chinese medicine and Western medicine, existing *E. sinica* populations and potential suitable areas are critical for its future protection and reforestation. We propose the following conservation management strategies for *E. sinica* in Inner Mongolia based on our findings.

Conservation methods such as in situ conservation, near-field conservation, ex situ conservation, and germplasm conservation have been gradually applied to preserve plant species [63]. In situ protection should be strengthened in regions where wild *E. sinica* exists, and protection-relevant policies and measures to prohibit the destruction of wild *E. sinica* should be issued. Furthermore, reasonable restoration project planning should be encouraged in the predicted potential suitable habitat areas. The majority of important suitable habitat for *E. sinica* in our study was located in the Horqin, Wulanchabu, Ordos and Ulan Butong grasslands. These regions account for 35% of Inner Mongolia and should be given top priority for restoration. Introduction and cultivation of wild *E. sinica* are important ways to restore this plant, and the government can increase farmers' enthusiasm for planting *E. sinica* through policies and economic incentives. Meanwhile, *E. sinica* cultivation should be based on imitation of wild cultivation mode [82]. Taking other species into account when conducting *Ephedra* resource protection, such as *Haloxylon ammodendron*, *Hippophae rhamnoides* Linn, and

*Glycyrrhiza uralensis* Fisch, could reduce environmental vulnerability and aid in the survival of *E. sinica* [35].

## 5. Conclusions

To analyze the geospatial pattern of ephedrine (E) and pseudoephedrine (PE) contents in *E. sinica*, we used a comprehensive geospatial quality model. The model results showed that *E. sinica* with high-quality E content was mainly distributed in the Horqin, Ulan Butong and Wulanchabu grasslands, while *E. sinica* with high-quality PE content was found primarily in the Ordos, Wulanchabu and Ulan Butong grasslands. At the same time, the key environmental factors affecting *E. sinica* were individuated through the fitted MaxEnt model. Climate, topographic and human activity variables had the greatest influence on *E. sinica*. The projected geospatial pattern and modeled habitat suitability for *E. sinica* are good references for sustainable use and conservation strategies.

## Supporting information

**S1 Data. Occurrence records of E.** sinica.
(XLS)

**S2 Data. Sample sites with two ephedrine-type alkaloids.**
(XLS)

## Author Contributions

**Conceptualization:** Longfei Guo.

**Data curation:** Longfei Guo, Ping He.

**Formal analysis:** Longfei Guo, Yu Gao, Xiaoting Zhang, Tongtong Huo.

**Funding acquisition:** Fanyun Meng.

**Investigation:** Ping He.

**Methodology:** Longfei Guo.

**Project administration:** Fanyun Meng.

**Software:** Longfei Guo, Ping He, Yuan He.

**Supervision:** Ping He, Yuan He, Yu Gao, Xiaoting Zhang, Tongtong Huo, Fanyun Meng.

**Validation:** Longfei Guo.

**Visualization:** Longfei Guo, Yuan He.

**Writing – original draft:** Longfei Guo.

**Writing – review & editing:** Longfei Guo, Cheng Peng.

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
