## [Decision Letter · Decision Letter 0]

17 Aug 2022

PONE-D-22-06770Predicting the comprehensive geospatial pattern of two ephedrine-type alkaloids for E. sinica in Inner MongoliaPLOS ONE

Dear Dr. Meng,

Thank you for submitting your manuscript to PLOS ONE. After careful consideration, we feel that it has merit but does not fully meet PLOS ONE’s publication criteria as it currently stands. Therefore, we invite you to submit a revised version of the manuscript that addresses the points raised during the review process.

Two out of three referees found merit in this manuscript, although a thorough revision is highly recommended. Specifically, the overall writing should be improved, also adding more background and case studies in the introduction and discussing more comprehensively the most important environmental factors emerging in the modelling exercise. Importantly, some major flaws in the modelling phase must be fixed, as e.g. the choice of the cross-validation approach and of the evaluation metrics.

We look forward to receiving your revised manuscript.

Kind regards,

Mirko Di Febbraro

Academic Editor

PLOS ONE

https://journals.plos.org/plosone/s/file?id=ba62/PLOSOne_formatting_sample_title_authors_affiliations.pdf".

“Funding for our research was provided by the National Natural Science Foundation of China (Grant No. 81072999).”

“We wish to express our gratitude to all the authors of this paper, all of whom provide useful feedback on an earlier version of the manuscript. Funding for our research was provided by the National Natural Science Foundation of China (Grant No. 81072999).”

“Funding for our research was provided by the National Natural Science Foundation of China (Grant No. 81072999).”

“The authors declare that they have no known competing financial interests or personal relationships that could have appeared to influence the work reported in this paper.”

6. We note that [Figures 1, 4, 6 & 7] in your submission contain [map/satellite] images which may be copyrighted. All PLOS content is published under the Creative Commons Attribution License (CC BY 4.0), which means that the manuscript, images, and Supporting Information files will be freely available online, and any third party is permitted to access, download, copy, distribute, and use these materials in any way, even commercially, with proper attribution. For these reasons, we cannot publish previously copyrighted maps or satellite images created using proprietary data, such as Google software (Google Maps, Street View, and Earth). For more information, see our copyright guidelines: http://journals.plos.org/plosone/s/licenses-and-copyright.

a. You may seek permission from the original copyright holder of [Figures 1, 4, 6 & 7] to publish the content specifically under the CC BY 4.0 license. 

Please upload the completed Content Permission Form or other proof of granted permissions as an """"Other"""" file with your submission.

Natural Earth (public domain): http://www.naturalearthdata.com/.

Reviewers' comments:

Reviewer's Responses to Questions

**Comments to the Author**

1. Is the manuscript technically sound, and do the data support the conclusions?

Reviewer #1: Partly

Reviewer #2: Partly

Reviewer #3: Partly

2. Has the statistical analysis been performed appropriately and rigorously? 

Reviewer #1: No

Reviewer #2: No

Reviewer #3: Yes

3. Have the authors made all data underlying the findings in their manuscript fully available?

Reviewer #1: Yes

Reviewer #2: Yes

Reviewer #3: No

4. Is the manuscript presented in an intelligible fashion and written in standard English?

Reviewer #1: Yes

Reviewer #2: No

Reviewer #3: Yes

5. Review Comments to the Author

Reviewer #1: The MS Predicting the comprehensive geospatial pattern of two ephedrine-type alkaloids for E.sinicain Inner Mongolia. Even though plant conservation is an important concern in a world that is changing, this MS cannot improve the conservation strategies for the researched plant in its current state. This MS is now a modeling exercise rather than a scientific study.

This MS cannot be approved for publishing unless substantial research and recommendations for habitat protection and conservation planning are included.

As stated by the context, indiscriminate destruction and grassland degradation pose major risks to species, and merely forecasting the species' probable range is insufficient, as these are in dire need of conservation.

It is beneficial to have distribution maps and predicted geographic trends, but it is unclear, and the author fails to address how to apply this knowledge for species conservation.

One of the ideas is that as authors are eager to predict the distribution of species, one of the ideas is that they evaluate if ensemble modelling is useful for optimising the predictive performance of a species distribution model in this instance.

Furthermore, authors should additionally incorporate other limiting factors, such as human pressure (e.g., human impact index) and other habitat disturbances that might further constrain the niche of the species into the model. The authors should study various local-scale techniques, such as PAB (propagule pressure, abiotic and biotic). Authors should utilize future possibilities as well. There is the potential for enhancing the models using variables such as "future land use and cover change scenarios."

Reviewer #2: Dear Authors,

I’ve been glad to review your manuscript entitled “Predicting the comprehensive geospatial pattern of two ephedrine-type alkaloids for E. sinica in Inner Mongolia”. I think the idea of coupling the predictions of habitat suitability for E. sinica resulting from Maxent with a spatial interpolation model focusing on the content of its two ephedrine-type metabolites may indeed allow you to provide guidance for a smart and conservative utilization of this plant. However, I have some major concerns in the way the Maxent model was fitted and evaluated, in the way you described the model building steps in your Methods section, as well as about some parts of the Results.

Pease find below my major comments, along with a set of minor comments aiming to aid you improving the clearness and conciseness of your manuscript.

Sincerely,

Francesco Cerasoli

Major comments

Introduction

L. 78: “High-quality species niches are different from species niches” is an incorrect formulation in my opinion. The niche of a species, intended as the Hutchinsonian multivariate fundamental niche, is one. What may differ among the different areas where a species is present is the degree to which the values of the abiotic factors characterizing those areas are close to the optimal range of values along the relevant axes of this multivariate niche space. Thus, I suggest replacing your statement with something like “areas hosting suitable conditions for high-quality populations of a pharmacological plant species (i.e. populations whose individuals show high content of the plant’s medicinal components) may not completely match areas where the species actually occurs”.

Methods

1. You should specify the resolution (e.g. 30 arc-seconds) of the different sets of environmental variables you initially considered.

2. You should specify which link function (I suppose Gaussian) you used to model the content of E and PE as a function the variables selected in the stepwise process.

3. You relied on the high AUC value your Maxent model attained on the test data to state that the corresponding model predictions are highly reliable. Nonetheless, lots of papers on ENMs/SDMs in recent years have shown that high AUC scores could be also artifacts relating to spatial autocorrelation between the training and test datasets, as well as depending on the relative extent of occurrence of the target species across the study area (see for instance Jiménez-Valverde et al., 2013; Roberts et al., 2017; Cerasoli et al., 2022). As you state that you used the default splitting criterium of the Maxent program (i.e. “Crossvalidate”), you could have ended up with test data being spatially (and likely environmentally) close to the training ones, which in turn could have inflated the AUC score.

Thus, to make your model even more reliable, I suggest you either:

o Use the ‘enmeval’ R package (Kass et al., 2022). This would allow you to define a spatially and/or environmentally clustered cross-validation structure which would reduce the risk of spatial autocorrelation between training and test data, and then to find the combination of parameters (i.e. value regularization multiplier, feature classes) which most reduces the difference between the AUC computed on training data and that computed on test data (i.e. reducing model overfitting).

o Add an additional model evaluation step using the Boyce index, which is more suited for presence-background algorithms as Maxent (Hirzel et al., 2006; Cerasoli et al., 2021).

Results

L. 261: “the linear relationship between the dependent variable and independent variable was obvious”…which independent variable? From the β parameters it seems that aspect variations have a poor influence on the E content (βaspect = 0.001). The same “criticism” applies to the Discussion section, L. 353-356, where you highlight the presumed importance of topographic factors on E content.

L. 330-332, and L. 336-338: According to the relative contribution scores you reported, precipitation seasonality (bio15) was the third most influential variable, not precipitation of the driest quarter (bio17). Please correct the corresponding parts of the Discussion.

Minor (Text-related) comments

Title: You should report the extended binomial name Ephedra sinica in the title.

L. 29: I would say “highly suitable areas for E. sinica” rather than “high-quality suitable area of E. sinica”.

L. 30: I would add “these” before “Two ephedrine-type alkaloids”.

L. 52: “Ecological environment” is a bit weird definition, consider replacing with “natural environments”. Here and in the other parts of the manuscript where you used this expression.

L. 56: “and important environmental factors” is maybe too generic; consider replacing with something like “and to assess the factors shaping environmental suitability for these latter”.

L. 63: Replace “Pepnia rockii” with “Peonia rockii”.

L. 66: In the Ecological Niche Modelling (ENM) literature, the “potential” adjective is mostly associated to the noun “distribution”. Indeed, ENM algorithms usually take advantage of occurrence data, usually depicting only a portion of the realized niche of the target species, to estimate the potential distribution of the species across the landscape (Peterson & Soberón, 2012). Thus, I suggest replacing “potential ecological niches” with “potential distribution”.

L. 92: I suggest replacing “this plant results in” with “the medicinal components of this plant favour”.

L. 101: I suggest replacing “etc.” with “along with other beneficial effects”.

L. 104: “ephedrine-type alkaloids are affected by different species and spatial locations” is not so clear; you may change with something like “ephedrine-type alkaloids differ in content and quality among species and geographical locations”.

L. 107: “ecological environmental improvements” is a bit weird formulation. Consider replacing by “ecological enhancement”.

L. 110: Add blank spaces between the single words in “useEand”.

L. 111: Consider replacing “of E. sinica to analyze…” with “of E. sinica, by analyzing…”.

L. 115: Add “potential” before “distribution”.

L. 123: I think it is better to use the extended binomial name of the target species within the sections’ headings and sub-headings.

L. 124: Replace “and E. sinica” with “this plant/species”.

L. 131-132: To avoid repeating the word “grassland” too many times, you could simply state “The main grassland areas of Inner Mongolia are Hulun Buir, Uragai, Xilingol…”.

L. 138: “improve the credibility of the model.” is too generic, you could state “to reduce the potential detrimental effect of spatial autocorrelation in the training dataset” or simply “to avoid redundancy in the information used to fit the model”.

L. 142: Use the plural form “sites” as you have several sampling points.

L.145: As the selection of predictors is not the only step affecting the results of ENMs, consider replacing “determines” with “contributes to determine”.

L. 147: “as the environmental data” is not very clear in this context. You could replace it with something “to individuate the variables being most informative for predicting…”.

L. 153: Replace “World Climate Database” with “WorldClim database”.

L. 157: I suggest replacing the link you provide for the Harmonized World Soil Database with this latter https://www.fao.org/soils-portal/data-hub/soil-maps-and-databases/harmonized-world-soil-database-v12/en which directly leads to the database portal.

L. 158: For more conciseness, consider replacing “from elevation variables using the spatial analysis function of ArcGIS (ESRI, Redlands, CA, USA), and the elevation data were obtained from WorldClim” with “from elevation data obtained from WorldClim, using the spatial analyst tool of ArcGIS software (ESRI, Redlands, CA, USA)”.

L. 161: Consider replacing “due to the multicollinearity of environmental variables” with “in case of multicollinearity among the environmental variables”.

L. 163-164: To avoid redundancy in this sentence, you may delete “to screen the environmental variables by group (Guo et al., 2021). We used r”

Caption of Table 1: add something like “, along with the respective relative contribution scores from the fitted Maxent model” at the end of the sentence.

Caption of Figure 2: Please report in the caption the extended forms of the abbreviations used in the diagram. I saw you reported them in the Methods subsection entitled “Comprehensive geospatial quality model”, yet the reader should be able to understand the Figure even without referring to the main text. Moreover, in the diagram of Figure 2 you should correct “envionmental” to “environmental”, and you may remove “result” after “habitat suitability” and “materials quality”.

L. 179: You should add some reference after “one of the most successful SDMs in predicting species distribution”, for instance Norberg, A., Abrego, N., Blanchet, F. G., Adler, F. R., Anderson, B. J., Anttila, J., ... & Ovaskainen, O. (2019). A comprehensive evaluation of predictive performance of 33 species distribution models at species and community levels. Ecological monographs, 89(3), e01370.

L. 181: Consider replacing “to project the distribution of E. sinica” with “to project the potential distribution of E. sinica across Inner Mongolia”. Moreover, to avoid redundancy, consider replacing “For the location point data, 75% of the data were selected as a training model, and the remaining…” with “75% of the occurrence data were selected to fit the model, and the remaining…”.

L. 185: When at the beginning of a period, use the extended binomial name Ephedra sinica.

L. 188: AUC values can be also lower than 0.5, please correct.

L. 189: I would say “a value of 0.5 indicates that model predictions are not better than random ones”, rather than” a value of 0.5 indicates that a model result is random”.

L.193: Consider “potential species distribution map was grouped into four classes” with “predicted suitability values were grouped into four classes”.

L. 219: “ is the result of the two ephedrine-type alkaloids in E. sinica in the th evaluation unit” is not very clear, consider replacing with “ is the output of the comprehensive geospatial quality model for the two ephedrine-type alkaloids of E. sinica in the th evaluation unit”.

L. 224: Consider using this formulation “the Maxent-based species distribution model (SDM) and the GLM-based geospatial quality model”.

L. 230-231: remove the dots within the extent measures of the ‘High’, ‘Medium’ and ‘Low’ suitability areas, as they may seem representing the decimal separator.

L. 242: bio15 is the coefficient of variation of precipitation, please correct. By the way, since bio15 is a ratio, it as no associated units of measurements so you should remove the “(mm)” indication in Table 1 while you can indicate “(%)”.

L. 244-246: “A variable response curve shows how each environmental variable affects a MaxEnt prediction and logistic predictions (Li et al., 2020). Hence, we used the response curve to obtain the parameters of the main influencing factors (Fig. 5)” is a bit weird sentence…consider replacing with something like “partial response curves reported in Fig. 5 show how Maxent predictions of habitat suitability for E. sinica changed along the gradient of the most influencing variables”.

L. 249: Here again, use % as “unit of measurement” for bio15.

L. 246-250: You should specific here which was the threshold used to identify the suitable range along the partial response curve. Was it the lower end of the “Medium” suitability interval? Please clarify.

L. 255-256: These lines should be moved to the Methods section describing the GLM used to spatially model E and PE content. In that section, you should also specify that the response variables of the model were the content of E and PE.

L. 256: add a reference for the SPSS software, or at least the version used.

L. 263-264: To avoid redundancy in this sentence, you may simply state “The spatial distribution of the E content in the different regions, modelled as a function of bio2 and aspect, is shown in Fig. 6a”.

L. 268: “The geospatial quality model of PE content and the environmental variables was…” is a bit weird sentence… consider replacing with “The geospatial quality model of PE content was represented by the equation…”.

L. 271-272: This is an “obvious” repetition of L. 260-262…please take better care of English syntax and avoid repeating entire sentences.

L. 273-274: Same comment than for L. 268…try to merge the two sentences into one to avoid unnecessary repetitions.

L. 278-283: As I commented above, the relationship between aspect variations and E content seems quite feeble…I would say that climate, and particularly temperature patterns, is apparently the main factor determining the distribution of E and PE content.

Figures 6 and 7: For better clarity, I suggest changing the legend titles to “E/PE content across Inner Mongolia” within the (a) panels and “E/PE content in important suitable habitats for E. sinica” within the (b) panels.

L.285: For better clarity, consider replacing “restricting the extent of two ephedrine-type alkaloids” with “to restrict the predictions of E and PE contents to areas with high suitability for E. sinica”.

L. 286-288: To be clearer and more concise, consider replacing “According to the results of the CGQ model of E in E. sinica, the comprehensive spatial pattern was predicted. Here, we defined an E content greater than 0.8% as high quality, and E. sinica with high-quality E content was mainly distributed…” with something like “According to the CGQ model, areas with predicted high-quality E content (i.e. > 0.8%) were mainly distributed…”

L. 292: Replace “of ephedra” with “of “E. sinica”.

L. 293-294: Here again, try to be more concise in your sentences. You can simply state “the predicted E content in important suitable areas for E. sinica was higher in the Horqin and Ulan Buh grasslands, while lower in the western Ordos grassland.”

L. 295-297: Try to avoid these repetitions of entire sentences…You can simply state that “areas with high-quality PE content (i.e. > 0.6%) resulting from the CGQ model mainly spanned…”

L. 305: If you know any previous work reporting these negative human impacts on E. sinica, put the corresponding reference here.

L. 309: the sentence “to predict geospatial pattern of two ephedrine-type alkaloids” is not very clear. You may change it with something like “to predict, across Inner Mongolia, the content of two ephedrine-type alkaloids characterising E. sinica”, here and throughout the manuscript.

L. 319-320: To avoid unnecessary repetitions, I suggest changing this sentence into “…various ecological factors, primarily climate, topography and soil properties”.

L. 323: Replace “bioclimate” with “bioclimatic”.

L. 326: which “cooccurring species”? please clarify.

L. 333: For a more appropriate English syntax, replace “have been buried in the soil for many years” with “persist in the soil for many years”.

L.373: Consider replacing “a sporadic distribution” with either “a fragmented distribution” or “sporadic occurrences”.

L. 384: You may delete “showed a trend that”.

L. 387: Consider replacing “such as the high E content” with “exemplified by the high E content”.

L. 388: Add “low” after “the E contents were”.

L. 425: For better clarity, consider replacing “were predicted in our results” with “were individuated through the fitted Maxent model”.

References:

Cerasoli, F, Besnard, A, Marchand, M-A, D'Alessandro, P, Iannella, M, Biondi, M. (2021) Determinants of habitat suitability models transferability across geographically disjunct populations: Insights from Vipera ursinii ursinii. Ecology and Evolution; 11: 3991– 4011. https://doi.org/10.1002/ece3.7294

Cerasoli F, D’Alessandro P, Biondi M. (2022) Fine-Tuned Ecological Niche Models Unveil Climatic Suitability and Association with Vegetation Groups for Selected Chaetocnema Species in South Africa (Coleoptera: Chrysomelidae). Diversity; 14(2):100. https://doi.org/10.3390/d14020100

Hirzel, A. H., Le Lay, G., Helfer, V., Randin, C., & Guisan, A. (2006). Evaluating the ability of habitat suitability models to predict species presences. Ecological Modelling, 199(2), 142–152.

https://doi. org/10.1016/j.ecolm odel.2006.05.017

Jiménez-Valverde, A., Acevedo, P., Barbosa, A. M., Lobo, J. M., & Real, R. (2013). Discrimination capacity in species distribution models depends on the representativeness of the environmental domain. Global Ecology and Biogeography, 22(4), 508–516.https://doi.org/10.1111/geb.12007

Kass, J. M., Muscarella, R., Galante, P. J., Bohl, C. L., Pinilla‐Buitrago, G. E., Boria, R. A., ... & Anderson, R. P. (2022) ENMeval 2.0: Redesigned for customizable and reproducible modeling of species’ niches and distributions. Methods in Ecology and Evolution, 12(9)

Roberts, D. R., Bahn, V., Ciuti, S., Boyce, M. S., Elith, J., Guillera-Arroita, G., Hauenstein, S., Lahoz-Monfort, J. J., Schröder, B., Thuiller, W., Warton, D. I., Wintle, B. A., Hartig, F., & Dormann, C. F. (2017). Cross-validation strategies for data with temporal, spatial, hierarchical, or phylogenetic structure. Ecography, 40(8), 913–929.https://doi.org/10.1111/ecog.02881

Reviewer #3: it is interesting to predicte the comprehensive geospatial pattern of secondary metabolites in plants of a certain species. this article is an good case study in this area. however, i suppose the authors may do something to help the readeers to better understand the work.

1. to add some background and case studies about the comprehensive quality model to determine the geospatial distribution of secondary metabolites in plants of a certain species.

2. to add the original data of E and PE content in E. sinica

3. to discuss more about how enviromental factors effect the E and PE content in E. sinica.

4. to improve the writing. I can understand the writhing, but I cannot say it is good and profesional. for instance, in the title, the name of the plant genus should not be just one letter. and what the unit of E and PE content in Fig 6-7?

6. PLOS authors have the option to publish the peer review history of their article (what does this mean?). If published, this will include your full peer review and any attached files.

Reviewer #1: No

Reviewer #2: **Yes: **Francesco Cerasoli

Reviewer #3: No

---

## [Author Response · Author response to Decision Letter 0]

6 Nov 2022

Enter any comments that you would like to send to the journal office. These comments will not appear in the submission or published article, if accepted.

If this manuscript has been returned to you with a request for corrections or information, provide a response here as directed in the email you received from the journal staff.

---

## [Decision Letter · Decision Letter 1]

25 Jan 2023

PONE-D-22-06770R1Predicting the comprehensive geospatial pattern of two ephedrine-type alkaloids for Ephedra sinica in Inner MongoliaPLOS ONE

Dear Dr. Meng,

Thank you for submitting your manuscript to PLOS ONE. After careful consideration, we feel that it has merit but does not fully meet PLOS ONE’s publication criteria as it currently stands. Therefore, we invite you to submit a revised version of the manuscript that addresses the points raised during the review process. The manuscript still suffers from major flaws especially about the written text. The referee provided a very detailed list of comments and suggestions to improve the manuscript readabillity. I strongly encourage the authors to take advantage of such comments to provide a markedly improved version of the manuscript.

We look forward to receiving your revised manuscript.

Kind regards,

Mirko Di Febbraro

Academic Editor

PLOS ONE

Reviewers' comments:

Reviewer's Responses to Questions

**Comments to the Author**

1. If the authors have adequately addressed your comments raised in a previous round of review and you feel that this manuscript is now acceptable for publication, you may indicate that here to bypass the “Comments to the Author” section, enter your conflict of interest statement in the “Confidential to Editor” section, and submit your "Accept" recommendation.

Reviewer #2: (No Response)

2. Is the manuscript technically sound, and do the data support the conclusions?

Reviewer #2: Yes

3. Has the statistical analysis been performed appropriately and rigorously? 

Reviewer #2: Yes

4. Have the authors made all data underlying the findings in their manuscript fully available?

Reviewer #2: Yes

5. Is the manuscript presented in an intelligible fashion and written in standard English?

Reviewer #2: No

6. Review Comments to the Author

Reviewer #2: Dear Authors,

I reviewed the updated version of your manuscript, now entitled “Predicting the comprehensive geospatial pattern of two ephedrine-type alkaloids for Ephedra sinica in Inner Mongolia”. Thank you for your point-by-point response letter and for your review effort.

I found the manuscript improved compared to the previous version. Nonetheless, I still find it is generally poorly written and really messy in some sections. You should make additional efforts to make your English wording clear, easy to read and syntactically correct, before your article could be published.

Please find my detailed comments in the attached file; I hope they will help you to ameliorate your manuscript.

As an advice for the future, please note that most reviewers would have not spent so much time in suggesting how to ameliorate single sentences and correct so many grammatical and syntactical mistakes…most of them would have likely rejected the manuscript just because it was unclear and incorrectly written. Thus, you should take care not only of the analytical procedures and corresponding results, but also of the way you present them.

Sincerely,

Francesco Cerasoli

7. PLOS authors have the option to publish the peer review history of their article (what does this mean?). If published, this will include your full peer review and any attached files.

Reviewer #2: **Yes: **Francesco Cerasoli

---

## [Author Response · Author response to Decision Letter 1]

4 Mar 2023

We are very grateful to the reviewer for the thorough review, positive feedback, and constructive comments. We have conducted profound reflection and revised the manuscript, adjusted the logic of the article, and polished our manuscript through language editing services to correct grammar and grammatical errors. Changes have been made in the revised manuscript based on the comments provided. Point-by-point responses are listed as follows.

Abstract: the period starting from L. 27 has several syntactical and grammatical errors, and it is too detailed about the optimization of model parameters. Consider replacing it with something like: “First, we used the MaxEnt software to map suitable habitats for E. sinica in inner Mongolia, based on occurrence data and a set of variables related to climate, topography and human impact. MaxEnt parametrization was optimized by evaluating alternative combinations of feature classes and values of the regularization multiplier. Secondly, a geospatial quality model was fitted to relate E and PE content to the same environmental variables and to predict their spatial patterns across the study area. Outputs from the two models were finally combined to map areas predicted to host both suitable conditions for E. sinica and high alkaloid content. Our results indicate that high-quality E content of E. sinica was mainly distributed in the Horqin grassland…”

Response: Thanks for the careful review. We have been replaced it (L. 27-38). 

Revision: “In this study, MaxEnt software was utilized to map suitable habitats for E. sinica in Inner Mongolia based on occurrence data and a set of variables related to climate, soil, topography and human impact. The model parametrization was optimized by evaluating alternative combinations of feature classes and values of the regularization multiplier. Second, a geospatial quality model was fitted to relate E and PE contents to the same environmental variables and to predict their spatial patterns across the study area. Outputs from the two models were finally coupled to map areas predicted to have both suitable conditions for E. sinica and high alkaloid content. Our results indicate that E. sinica with high-quality E content was mainly distributed in the Horqin, Ulan Butong and Wulanchabu grasslands. E. sinica with high-quality PE content was primarily found in the Ordos, Wulanchabu and Ulan Butong grasslands. This study provides scientific information for the protection and sustainable utilization of E. sinica. It can also help to control and prevent desertification in Inner Mongolia.”

L.20: To improve clarity, consider replacing “is a type of traditional Chinese medicine that has a high medicinal value and large demands.” with “is a shrubby plant widely used in traditional Chinese medicine due to high medicinal value and large demands”. 

Response: Thanks for the careful review. We changed the whole sentence (L. 21-22). 

Revision: “Ephedra sinica Stapf. is a shrubby plant widely used in traditional Chinese medicine due to its high level of medicinal value, thus, it is in high demand.” 

L. 26: To improve clarity, consider adding “content” after “E and PE”.

Response: We have been added it (L. 31). 

Revision: “Second, a geospatial quality model was fitted to relate E and PE contents to the same environmental variables and to predict their spatial patterns across the study area.”

L. 44: Please replace “relying” with “relied” in the sentence “and the majority of these studies relying on species distribution models”.

Response: We have deleted the sentence. 

L. 45-47: I suggest reversing the construction of this sentence to improve clarity. For instance, you could write: “Various SDM algorithms, including CLIMEX, DOMAIN, genetic algorithm for rule set production (GARP), BIOMOD-2 and maximum entropy (MaxEnt), have been used to assess ecological requirements and predict habitat suitability for several medicinal plant species”.

Response: We have replaced it (L. 68-71). 

Revision: “Various SDM algorithms, including CLIMEX, DOMAIN, the genetic algorithm for rule set production (GARP), BIOMOD-2 and maximum entropy (MaxEnt), have been used to assess ecological requirements and predict habitat suitability for medicinal plant species.”

L. 53: Please remove “the” before “MaxEnt”, here and throughout the manuscript unless you place words like “software”, “algorithm”, “model” after “MaxEnt”.

Response: We have removed “the” before “MaxEnt” throughout the manuscript (L. 76, 261). 

L. 60: Please replace “influencing” with “influence”.

Response: We have replaced it (L. 83).

L. 66: Please add “on” before “the suitable habitat” and before “the geospatial pattern”.

Response: We have changed the whole sentence (L. 87-89). 

Revision: “Thus, studies on high-quality suitable habitat for TCM plants should pay attention not only on the suitable habitat but also on the geospatial pattern of the effective medicinal components.”

L. 67-68: I suggest moving the phrase “We used a comprehensive quality model to determine the geospatial distribution of high-quality E. sinica” directly at the end of this paragraph (L.83), so that you start introducing the methods and scope of your study after having exemplified results from previous studies on medicinal plants. I also suggest adding “populations” after “E. sinica”.

Response: Thanks for the suggestion. But for the sake of the article as a whole, we deleted this paragraph.

L. 78: Please replace “develop” with “assess/estimate/analyze”. 

Response: We have changed the whole sentence (L. 98-100). 

Revision: “Furthermore, a study of habitat influence on the baicalin content in Scutellaria baicalensis Georgi revealed that climatic factors were important indicators of both the potential distribution and baicalin production of the plant.”

L. 80-82: “they found that climatic factors wouldn’t only be an indicator for identifying the potential distribution area, but also for forming some secondary metabolites such as baicalin” is a bit weird sentence…consider replacing it with something like “they found that climatic factors are useful indicators of both plants’ potential distribution and production of some secondary metabolites such as baicalin”. 

Response: We have changed the whole sentence (L. 98-100). 

Revision: “Furthermore, a study of habitat influence on the baicalin content in Scutellaria baicalensis Georgi revealed that climatic factors were important indicators of both the potential distribution and baicalin production of the plant.”

L. 86: Please replace the comma after “colds and obesity[20]” with a semicolon or a full stop. 

Response: We have replaced it (L. 49). 

Revision: “It is commonly used in TCM to treat a variety of ailments, including bronchitis, asthma, whooping cough, colds and obesity[5]. ”

L. 100: Please replace “overharvested” with “overharvesting”. 

Response: We have replaced it (L. 63). 

Revision: “However, the increasing international market demand for natural alkaloid contents has led to overharvesting of E. sinica.”

L. 105: Add “geospatial” before “patterns” and add “these” before “two”. 

Response: We have changed the sentence (L. 104). 

Revision: “To generate and accumulate effective components, the plant must be alive; thus, when simulating the geospatial patterns of these two ephedrine-type alkaloids, habitat suitability for E. sinica should be delimited first.”

L. 108: Consider replacing “the spatial pattern of two ephedrine-type alkaloids” with “the spatial patterns of E and PE content”. 

Response: We have changed the sentence (L. 107-108). 

Revision: “Then we used a geospatial quality model and ephedrine-type alkaloid content data to predict the spatial patterns of E and PE contents.”

L. 110: To improve clarity, consider replacing “habitat distribution and effective components” with “the distribution of suitable habitats for E. sinica and effective production of ephedrine-type alkaloids”. 

Response: We have changed it (L. 110-111). 

Revision: “The results will help determine which environmental factors influence the distribution of suitable habitats for E. sinica and the effective production of ephedrine-type alkaloids.”

L. 122: To improve syntactical correctness, consider replacing “…Ordos. While the western region is arid, with the Alxa desert.” with “…Ordos, while the western region is arid and includes the Alxa desert”. 

Response: We have changed it (L. 121-122). 

Revision: “The grasslands are mostly distributed in eastern and central Inner Mongolia, including Hulun Buir, Uragai, Xilingol, Horqin, Ulan Butong, Wulanchabu and Ordos, while the western region is arid and includes the Alxa desert.”

L. 123: To improve syntactical correctness and clearness, consider replacing “…and sample sites with two ephedrine-type alkaloids were obtained through published researchers” with “…and sample sites with content data for the two target two ephedrine-type alkaloids were obtained through published literature”. 

Response: Thanks for the suggestion. We have replaced it (L. 123-124). 

Revision: “Occurrence records of E. sinica and sample sites with content data for the two target ephedrine-type alkaloids were obtained through published literature.”

Table 1: Please replace “Isothermally” with “Isothermality”. 

Response: We have been fixed the description of bio3.

Revision:

Table 1

Environmental variables used or not used in model, along with the respective relative contribution scores from the fitted Maxent model 

Variable type Code(Unit) Description Variables used

in modeling Contribution (%)

Climatic variables bio1(℃) Annual mean air temperature √ 2.1

 bio2(℃) Mean diurnal temperature range (max. temp – min. temp) √ 0.8

 bio3 Isothermality (Bio2/Bio7) × 100 √ 4.3

 bio4(℃) Temperature seasonality 

 bio5(℃) Max temperature of warmest month 

 bio6(℃) Min temperature of coldest month 

 bio7(℃) Temperature annual range √ 26.6

 bio8(℃) Mean temperature of wettest quarter 

 bio9(℃) Mean temperature of driest quarter 

 bio10(℃) Mean temperature of warmest quarter 

 bio11(℃) Mean temperature of coldest quarter 

 bio12(mm) Annual precipitation √ 8.4

 bio13(mm) Precipitation of wettest month 

 bio14(mm) Precipitation of driest month 

 bio15(%) Coefficient of variation of precipitation √ 19.3

 bio16(mm) Precipitation of wettest quarter 

 bio17(mm) Precipitation of the driest quarter 

 bio18(mm) Precipitation of warmest quarter 

 bio19(mm) Precipitation of coldest quarter 

 srad(kJ·m-2·d-1) Solar radiation √ 10.9

 vapr(hPa) Vapour pressure √ 0.6

Soil variables soil Soil type √ 0.3

 clay1 Topsoil Clay Fraction (0 - 30cm) 

 clay2 Subsoil Clay Fraction (30 - 100cm) 

 sand1 Topsoil Sand Fraction (0 - 30cm) √ 0.7

 sand2 Subsoil Sand Fraction (30 - 100cm) 

 sq1 Nutrient availability √ 0.3

 sq2 Nutrient retention capacity 

 sq3 Rooting conditions √ 1.1

 sq4 Oxygen availability to roots 

 sq5 Excess salts √ 1.1

 sq6 Toxicity √ 0.2

 sq7 Workability (constraining field management) 

Topographical variables ele(m) Elevation above sea level √ 0.2

 slop(%) Slope 

 asp(degrees) Aspect √ 0.1

Human activity variables hf Human Footprint 

 den Population Density 

 ter Global Human Modification of Terrestrial Systems √ 23

L.167-169: To improve clearness and readability, I would change the period structure to something like “By coupling a species distribution model with a geospatial quality model (GQM), we developed a comprehensive geospatial quality model (CGQM) to predict the geospatial pattern of two ephedrine-type alkaloids in E. sinica (Fig. 2). 

Response: We have changed it (L. 167-169). 

Revision: “By coupling a species distribution model with a geospatial quality model (GQM), we developed a comprehensive geospatial quality model (CGQM) to predict the geospatial pattern of two ephedrine-type alkaloids in E. sinica (Fig. 2).”

Figure 2: “Envionment” instead of “Environmental” is still present in the “Geospatial quality model” panel; please fix it. Moreover, consider adding “implemented” before “methodological approaches” in the caption. Finally, to improve clearness of the Figure content, I also suggest: 

- replacing “Spatial distribution of medicinal quality materials (GQM)” with “areas with high predicted alkaloid content”, also modifying the heading to “Geospatial Quality Model (GQM)”. 

- replacing “High quality suitable habitat of major content” with “highly suitable areas with high predicted alkaloid content” in the “Comprehensive Geospatial Quality Model (CGQM)” panel. 

Response: We have corrected the Figure 2.

Revision:

Fig. 2. Schematic representation of the methodological approaches.

L. 185: when writing “omission rates (E = 0.05)”, did you mean that you selected 0.05 as threshold value for omission rate thus considering only those models that did not exceed this value? Please clarify. 

Response: We have changed it (L. 184). 

Revision: “Model selection was based on (i) omission rates (E ≤ 0.05), which represent the proportion of incorrectly predicted test records by the model, and (ii) Delta Akaike Information Criterion (dAIC < 2), which indicates the models with the best trade-offs between data fitting and complexity.”

L. 189-191: This paragraph is poorly written. Consider adjusting to something like “Finally, the selected MaxEnt model was fitted using linear and threshold features (i.e. FC = lt), with RM = 3.5. The maximum number of iterations was set to 1000, logistic output format was set to logistic, and 10 replicated runs of cross-validation were used to reduce the uncertainty of the model.” 

Response: We have changed it (L. 188-192). 

Revision: “Finally, the selected MaxEnt model was fitted using linear and threshold features (i.e., FC = lt), with RM = 3.5. The maximum number of iterations was set to 1000, the logistic output format was set to logistic, and 10 replicated runs of cross-validation were used to reduce the uncertainty of the model.”

L. 195: Add the final “l” to “mode

Response: We have corrected it (L. 197).

L. 196: Please change “when it below 0.5 indicating…” to “with values lower than 0.5 indicating…” 

Response: We have changed it (L. 198). 

Revision: “AUC values range from 0 to 1, with values lower than 0.5 indicating that model predictions are not better than random ones and 1 representing perfect discrimination.”

L. 200-201: Consider moving this sentence about the split of predicted suitability into discrete classes to L. 193, so that it would be better connected to the description of MaxEnt output. 

Response: We have moved it (L. 194-195). 

Revision: “The final predicted suitability values were grouped via natural breaks into four classes of High, Medium, Low and Unsuitable.”

L.212-213: As in the previous version you stated that stepwise regression was used for model refinement (i.e. variable selection), consider changing this part to something like “We used SPSS (IBM SPSS Statistics 26; https://www.ibm.com/cn-zh/analytics/spss-statistics-software) to perform a stepwise regression for refining the initial GLM, thus maintaining only the informative predictors”. 

Response: We have changed the sentence (L. 213-215). 

Revision: “We used SPSS (IBM SPSS Statistics 26; https://www.ibm.com/cn-zh/analytics/spss-statistics-software) to perform a stepwise regression to refine the initial GLM, thus maintaining only the informative predictors.”

L. 234: Following my previous comment, change “GLM-based” with “regression-based”. 

Response: We have changed the sentence (L. 239). 

Revision: “The MaxEnt-based species distribution model (SDM) and regression-based geospatial quality model served as the foundation for the comprehensive geospatial quality model.”

L. 238-239: As the AIC value is not very informative by itself, you may remove it and change the sentence to “and AUC value of 0.960, indicating that the model performed well and was reliable in discriminating suitable areas for E. sinica”. 

Response: We have changed it (L. 243-245). 

Revision: “and AUC value of 0.965, indicating that the model performed well and was reliable in discriminating suitable areas for E. sinica.”

L.240-245: I suggest you to change “was” with “covered” when writing about the extent the different suitability classes spanned. Moreover, in explicating the extent of the different categories, avoid using XXX ˟ YYY km2 : you should either use the form XXX ˟ YYY km or directly enumerate the overall extent, such as ZZZ km2 . Please also change “‘High’ suitable habitat mainly distributed in…” with “highly suitable habitat was mainly distributed in…”. 

Response: We have changed them (L. 245-248). 

Revision: “The predicted suitability was divided into four levels: high suitability (> 0.55) was 161400 km2, medium suitability (0.34-0.56) was 260400 km2, low suitability (0.14-0.34) was 285700 km2, and the remaining area was unsuitable (<0.13) (Fig. 3). High-suitability habitat was mainly distributed in the Horqin, Wulanchabu and Ordos grasslands.”

L. 250-254: Here you state that the most influential variable was bio7, but Fig. 4a shows the partial response curve for bio17…please fix this. 

Response: We apologize for the mistake and fixed it (L. 245-248). 

Revision: 

Fig. 4. Response curves for important environmental predictors in the species distribution model for E. sinica. 

L. 256: You should not start a sentence with “And” after a full stop. Use instead something like “Further”, “Moreover”, “Additionally”. 

Response: We changed “And” with “Moreover” (L. 261).

Revision: “Moreover, our model results indicated that the suitable range for bio7...”

L. 257-260: I don’t think it is very informative to specify with such precision the range of values the partial response curves span. One reason is that this range could be influenced by the clamping procedure used within the MaxEnt software to extend the response curves beyond the range of the training values…for instance, how could the optimal value for ter be “1～1.9” if you state that it “ranged from 0.38 to 1”. I suggest you focus instead on the trends resulting from the curves (e.g. increasing suitability for E. sinica with higher values of ter and bio15) and on the range of values (not the precise values which are difficult to assess from the curves) corresponding to the highest predicted suitability and/or to abrupt changes in the trend of the focal curve. 

Response: We changed the sentence (L. 261-267).

Revision: “Moreover, our model results indicated that the suitable range for bio7 was approximately 45 °C to 50 °C, and the optimal value of bio7 was approximately 46 °C. In addition, bio15 ranged from 105% to 130% with an optimal value at approximately 115%; srad ranged from 15000 kJ·m-2·d-1 to 16500 kJ·m-2·d-1, with the highest predicted suitability value at 15500 kJ·m-2·d-1; bio 12 ranged from 200 mm to 450 mm with an optimal elevation at approximately 270 mm. Furthermore, as the value of ter increased above 0.1, an increase in the probability of presence was observed.”

L. 270: Please change “This result showed that the model had a remarkable effect,” with something like “This result suggests that the model explained a not negligible portion of the spatial variability in E content”. A model has no effect…the model you used just estimate the portion of variability in the response variable which could be causally related to the retained predictors… 

Response: We changed the sentence (L. 277-278).

Revision: “This result suggests that the model explained a nonnegligible portion of the spatial variability in E content,”

L. 282: As for the previous comment, please replace “the model had a significant effect, and the result is reliable” with something like “the model has a noticeable explanatory power”. 

Response: We changed the sentence (L. 285-286).

Revision: “This result demonstrated that the model has noticeable explanatory power.”

L. 283: To improve clearness and syntactical correctness, please replace “and the spatial quality model result of PE in the study area shown in…” with something like “the projection of PE content across Inner Mongolia resulting from the corresponding geospatial quality model is shown in…” 

Response: We changed the sentence (L. 287-288).

Revision: “the projection of PE content across Inner Mongolia resulting from the corresponding geospatial quality model is shown in Fig. 6a.”

L. 290-293: The wording here is a bit blurry…consider replacing with something like “Climate, and particularly temperature patterns, is apparently the main factor determining the accumulation of E and PE in E. sinica samples because bio2 and bio8 emerged as the main variables influencing the spatial distribution of E and PE content”.

Response: We changed the sentence (L. 289-291).

Revision: “Climate, particularly temperature patterns, is apparently the main factor determining the accumulation of E and PE contents in E. sinica samples because bio2 and bio8 emerged as the main variables influencing the spatial distribution of E and PE contents.”

L. 298: To improve clearness and syntactical correctness, please replace “The high-quality E content in ISH was approximately 13.86 × 104 km2” with something like “the extent of ISH areas predicted to host high-quality E was approximately ZZZ km2 ”. 

Response: We changed it (L. 296).

Revision: “and they covered approximately 138600 km2.”

L. 314-315: Consider replacing “and the AUC value was 0.960, indicating excellent performance in evaluating the accuracy of the results. Further, we used a GLM to…” with something like “and the AUC value was 0.960, indicating excellent discrimination performance. Further, we used a stepwise regression-based model to…” 

Response: We changed the sentence (L. 312-313).

Revision: “This study used a MaxEnt model to simulate the potential distribution of E. sinica in Inner Mongolia, for which the AUC value was 0.960, indicating excellent discrimination performance. Furthermore, we used a stepwise regression-based model to predict, across Inner Mongolia…”

L. 318: Add “these” before “two models”. 

Response: We added it (L. 316).

Revision: “We obtained the final comprehensive geospatial distribution for two ephedrine-type alkaloids of E. sinica by coupling the results of these two models.”

L. 320-321: You should better explicit how your results would help to “promote the sustainable use and appropriate conservation of E. sinica in Inner Mongolia”, for instance adding something like “by indicating which areas potentially host higher E. sinica populations with ephedrine-type alkaloid content so as to make its harvesting more efficient”. If you do this, you could then start the following sentence with “Our results will also make an important contribution to the enhancement of medicinal value and protection of biodiversity in Inner Mongolia”. 

Response: We added it (L. 318-321).

Revision: “These findings will promote the sustainable use and appropriate conservation of E. sinica in Inner Mongolia by indicating which areas can potentially host larger E. sinica populations with high-quality ephedrine-type alkaloid content; these conditions can increase harvesting efficiency. Our results also make an important contribution to the enhancement of medicinal value and protection of biodiversity in Inner Mongolia.”

L. 326-327: Please remove this sentence as it is totally redundant with the previous one. 

Response: We removed it.

L. 333-334: The wording here is a bit awkward; consider changing to something like: “In addition, climate emerged as the main factor driving the distribution of Rosa arabica Crep. which frequently cooccurs with Ephedra species”. 

Response: We changed the sentence (L. 350-351).

Revision: “In addition, climate emerged as the main factor driving the distribution of Rosa arabica Crep., which frequently cooccurs with Ephedra species”

L. 335: Please add “for E. sinica” after “habitat suitability”. 

Response: We changed the sentence (L. 353).

Revision: “Similar to previous studies, our results indicated that climate-related predictors were the main variables affecting habitat suitability for E. sinica.”

L. 339: “The temperature factor provides E. sinica an comfortable environment to grow up…” is a bit awkward sentence; consider replacing with something like “Annual temperature range contributes to determine the degree to which E. sinica populations of Inner Mongolia experience comfortable thermal conditions to grow up…” 

Response: We changed the sentence (L. 357-359).

Revision: “Annual temperature range helps determine the appropriate thermal conditions that E. sinica populations in Inner Mongolia need to grow,”

L. 340: Please remove the word “factor” after precipitation…unless you’re referring to a specific precipitation-related variable it is uninformative. 

Response: We changed the sentence (L. 366).

L. 347: To avoid redundancy, consider this formulation: “…for the distribution of E. sinica, because it grows…” 

Response: We changed the sentence (L. 366).

Revision: “Furthermore, global human modification of terrestrial systems (23%) is an important variable for the distribution of E. sinica because E. sinica grows primarily in grasslands, where farming and animal husbandry activities and cultivation behavior change the land cover.”

L. 349-351: Here you state soil variables are also important, but this did not emerge from your MaxEnt model, so you should discuss why…for instance, you may hypothesize that soil properties are important drivers of E distribution at finer spatial scales than the regional scale of your research. 

Response: We changed the sentence (L. 367-370).

Revision: “However, soil variables, such as soil texture, vertical soil properties and root infiltration conditions, directly affect the survival and growth of perennial roots, thus indicating that soil properties are important drivers of E. sinica distribution at finer spatial scales.”

L. 352: “Secondary metabolite from plants is the unique sources…” is not so convincing. Pharmaceuticals, flavours etc. also derive from animal or mineral material. Please replace with “Secondary metabolites from plants are among the main sources…” 

Response: We changed the sentence (L. 371).

Revision: “Secondary metabolites from plants are among the main sources of pharmaceuticals, food additives, flavors, and other industrial materials.”

L. 357: Please correct to “Ephedrine-type alkaloids are the main secondary metabolites…” 

Response: We changed the sentence (L. 376-378).

Revision: “Ephedrine-type alkaloids, which are the main secondary metabolites of Ephedra, are important for estimating the quality of crude drugs and safe medication.”

L. 360: Please correct to “various mechanisms, which are regulated…” 

Response: We changed the sentence (L. 380-382).

Revision: “To cope with heat stress, plants implement various mechanisms that are regulated at the molecular level, which can improve plant heat stress tolerance and enable plants to thrive under heat stress.”

L. 361: You should expand a bit about these “genetic strategies” to make the concept clearer to a reader not experienced of plants physiological regulatory mechanisms. 

Response: After reviewing the literature, we found that there was no professional term to explain “genetic strategies”, so we changed the sentence (L. 384-386).

Revision: “To cope with heat stress, plants implement various mechanisms that are regulated at the molecular level, which can improve plant heat stress tolerance and enable plants to thrive under heat stress.”

L. 365: Break the sentence here: “…temperature treatments [70]; for instance, the concentrations of ergine, ergonovine…” 

Response: We changed the sentence (L. 384-385).

Revision: “Furthermore, different alkaloids exhibit different change trends when subjected to different temperature treatments [71]; for instance, the concentrations of ergine, ergonovine, and total ergot alkaloids are significantly higher when Festuca sinensis is maintained under short-term cold stress.”

L. 366: Please add something like “is maintained” or “is put” before “under short-term cold stress”. 

Response: We added “is maintained” before “under short-term cold stress” (L. 386).

Revision: “and total ergot alkaloids are significantly higher when Festuca sinensis is maintained under short-term cold stress.”

L. 367: Add ‘range’ after ‘mean diurnal temperature’. 

Response: We added it (L. 387).

Revision: “In our study, the mean diurnal temperature range was the key variable for the generation of ephedrine, and it was positively correlated with the change in E content”

L. 367-369: As here you’re citing for the first time in the Discussion the two ephedrine-type alkaloids, you may use their extended names to help the reader recall what you’re talking about, then continuing with their abbreviations in the subsequent paragraphs. 

Response: We have been corrected it (L. 387,389).

Revision: “In our study, the mean diurnal temperature range was the key variable for the generation of ephedrine (E), and it was positively correlated with the change in E content. When the temperature rose, the E content increased. The most important factor for the production of pseudoephedrine (PE) was the mean temperature of the wettest quarter, which was negatively correlated with PE content, so that the latter increased with lower temperature in the wettest season.”

L. 370: To ameliorate the wording, consider changing to something like “which was negatively correlated to PE content, so that the latter increases with lower temperature in the wettest season.” 

Response: We have been changed the sentence (L. 390-391).

Revision: “The most important factor for the production of pseudoephedrine (PE) was the mean temperature of the wettest quarter, which was negatively correlated with PE content, so that the latter increased with lower temperature in the wettest season.”

L. 378: Please change ‘species’ to ‘populations/variety’ if you’re specifically referring to E. sinica. 

Response: We have been changed ‘species’ to ‘populations’ (L. 398).

Revision: “The survival and competition potential of local populations of E. sinica should be evaluated carefully, and the destruction of local habitats and biodiversity should be avoided in the process of artificial cultivation”

L. 379: Consider changing “ecological environment” with “habitats”. 

Response: We have been changed “ecological environment” with “habitats” (L. 399).

Revision: “The survival and competition potential of local populations of E. sinica should be evaluated carefully, and the destruction of local habitats and biodiversity should be avoided in the process of artificial cultivation”

L. 384-388: Here you write “E. sincia” instead of “E. sinica”; please correct. 

Response: We apologize for the misspelling. We corrected it (L. 422-426).

Revision: “As a result of rapidly changing climate and human-caused ecological destruction, the distribution of E. sinica in Inner Mongolia has shifted from widespread to fragmented distribution in recent decades [78, 79]. Meanwhile, because of the high medicinal value of E. sinica, which is widely used in both traditional Chinese medicine and Western medicine, existing E. sinica populations and potential suitable areas are critical for its future protection and reforestation. We propose the following conservation management strategies for E. sinica in Inner Mongolia based on our findings.”

L. 384: Replace “have been” with “has been” as you’re referring to a singular subject, “E. sinica”. 

Response: We corrected it (L. 423).

Revision: “Meanwhile, because of the high medicinal value of E. sinica, which is widely used in both traditional Chinese medicine and Western medicine, existing E. sinica populations and potential suitable areas are critical for its future protection and reforestation. We propose the following conservation management strategies for E. sinica in Inner Mongolia based on our findings.”

L. 385: Add “populations” or “occurrence localities” after “existing E. sinica”. 

Response: We added it (L. 425).

Revision: “Meanwhile, because of the high medicinal value of E. sinica, which is widely used in both traditional Chinese medicine and Western medicine, existing E. sinica populations and potential suitable areas are critical for its future protection and reforestation. We propose the following conservation management strategies for E. sinica in Inner Mongolia based on our findings.”

L. 391: To avoid repeating “conservation” so many times, consider replacing “to achieve species conservation in recent years” with “to preserve plant species”. 

Response: We added it (L. 429-430).

Revision: “Conservation methods such as in situ conservation, near-field conservation, ex situ conservation, and germplasm conservation have been gradually applied to preserve plant species.”

L. 393-394: These two lines could be removed as you few lines later you specify the ISH areas which should be given restoration priority. 

Response: We removed the sentence to L. 432-433.

Revision: “In situ protection should be strengthened in regions where wild E. sinica exists, and protection-relevant policies and measures to prohibit the destruction of wild E. sinica should be issued. Furthermore, reasonable restoration project planning should be encouraged in the predicted potential suitable habitat areas. The majority of important suitable habitat for E. sinica in our study was located in the Horqin, Wulanchabu, Ordos and Ulan Butong grasslands.”

L. 396: Consider removing the “ISH covered ca. 42.18 × 104 km2” part, because repeating in the Discussion quantitative details you already reported in the Results is not so useful. 

Response: We removed the sentence (L. 434).

L. 399 and L. 405: You may remove the saxon genitive here as you never used it above when referring to E. sinica. 

Response: We removed the sentence (L. 438 and L. 402).

L. 425: Please change “need” to “needed” as here you’re referring to what you’ve done for your specific research. 

Response: We changed “need” to “needed” (L. 325).

Revision: “We needed to collect not only occurrence records of E. sinica but also sites with corresponding effective component data; however, collecting data on components is difficult because few studies have been done on the spatial patterns of the two ephedrine-type alkaloids.”

L. 429: To improve the wording, consider changing to “…the medicinal parts of the target plant species”. 

Response: We changed the sentence (L. 329).

Revision: “We believe that component data richness could be increased by using field work to collect the medicinal parts of the target plant species and by implementing additional laboratory work to measure the effective components if corresponding hardware devices are available.”

L. 430: “in future studies” is a bit redundant here, you may remove it. 

Response: We removed it (L. 331).

L. 433: To improve the wording, consider changing to “..environmental variables, plant distribution as well as metabolite contents”. 

Response: We changed the sentence (L. 333-334).

Revision: “and we need to intuitively understand the relationship between environmental variables, plant distribution and metabolite contents.”

L. 435-439: To improve the wording, consider changing to “…According to the chosen evaluation metrics, both models performed well, and they identified climate, human modifications and, secondarily, topography as key environmental factors affecting the distribution of E. sinica and the spatial pattern of alkaloids content. The fact that we used environmental predictors with approximately 1 Km2 resolution may facilitate extracting general guidelines on conservation and management of E. sinica in Inner Mongolia at a relatively fine spatial scale”. 

Response: We changed the sentence (L. 336-341).

Revision: “According to the chosen evaluation metrics, both models performed well, and they identified climate, human modifications and, secondarily, topography as key environmental factors affecting the distribution of E. sinica and the spatial pattern of alkaloid content. The fact that we used environmental predictors with approximately 1 km2 resolution may facilitate the development of general guidelines on the conservation and management of E. sinica in Inner Mongolia at a relatively fine spatial scale.”

L. 443: As for the start of the Discussion, I suggest using the extended name of E and PE here that you cite them for the first time in the Conclusions section. 

Response: We corrected it (L. 443).

Revision: “To analyze the geospatial pattern of ephedrine(E) and pseudoephedrine(PE) contents in E. sinica…”

L. 448: Add human modifications as a key predictor, as it emerged clearly within your updated Maxent model. 

Response: We corrected it (L. 448-449).

Revision: “Climate, topographic and human activity variables had the greatest influence on E. sinica.”

L. 449: Please replace “environmental variables” with “modelled habitat suitability”, as in its current form this sentence is unclear

Response: We replaced “environmental variables” with “modelled habitat suitability” (L. 450).

Revision: “The projected geospatial pattern and modeled habitat suitability for E. sinica are good references for sustainable use and conservation strategies.”

---

## [Decision Letter · Decision Letter 2]

21 Mar 2023

Predicting the comprehensive geospatial pattern of two ephedrine-type alkaloids for Ephedra sinica in Inner Mongolia

PONE-D-22-06770R2

Dear Dr. Meng,

We’re pleased to inform you that your manuscript has been judged scientifically suitable for publication and will be formally accepted for publication once it meets all outstanding technical requirements.

Kind regards,

Mirko Di Febbraro

Academic Editor

PLOS ONE

Additional Editor Comments (optional):

Reviewers' comments:

Reviewer's Responses to Questions

**Comments to the Author**

1. If the authors have adequately addressed your comments raised in a previous round of review and you feel that this manuscript is now acceptable for publication, you may indicate that here to bypass the “Comments to the Author” section, enter your conflict of interest statement in the “Confidential to Editor” section, and submit your "Accept" recommendation.

Reviewer #2: All comments have been addressed

2. Is the manuscript technically sound, and do the data support the conclusions?

Reviewer #2: Yes

3. Has the statistical analysis been performed appropriately and rigorously? 

Reviewer #2: Yes

4. Have the authors made all data underlying the findings in their manuscript fully available?

Reviewer #2: Yes

5. Is the manuscript presented in an intelligible fashion and written in standard English?

Reviewer #2: Yes

6. Review Comments to the Author

Reviewer #2: (No Response)

7. PLOS authors have the option to publish the peer review history of their article (what does this mean?). If published, this will include your full peer review and any attached files.

Reviewer #2: **Yes: **Francesco Cerasoli

---

## [Editor Report · Acceptance letter]

13 Apr 2023

PONE-D-22-06770R2 

Predicting the comprehensive geospatial pattern of two ephedrine-type alkaloids for *Ephedra sinica* in Inner Mongolia 

Dear Dr. Meng:

I'm pleased to inform you that your manuscript has been deemed suitable for publication in PLOS ONE. Congratulations! Your manuscript is now with our production department. 

Kind regards, 

on behalf of

Dr. Mirko Di Febbraro 

Academic Editor

PLOS ONE